# Wnt10b signaling regulates replication stress–induced chromosomal instability in human cancer

Alexander Haas[1], Friederike Wenz[1] , Janina Hattemer[2], Janine Wesslowski[3] , Gary Davidson[3] ,
Oksana Voloshanenko[4], Michael Boutros[4], Sergio P Acebron[2,5], Holger Bastians[1]

**Wnt signaling pathways are involved in various developmental and tissue maintenance functions, whereas deregulated Wnt signaling is closely linked to human cancer. Recent work revealed that loss of Wnt signaling impairs mitosis and causes abnormal microtubule growth at the mitotic spindle resulting in chromosome missegregation and aneuploidy, both of which are hallmarks of cancer cells exhibiting chromosomal instability (CIN). Here, we show that upon DNA replication stress, a condition typically associated with CIN, Wnt10b acts to prevent increased microtubule dynamics from the S phase until mitosis, thereby ensuring faithful chromosome segregation. Interestingly, replication stress–induced chromosomal breaks are also efficiently suppressed by Wnt10b. Thus, our results show that Wnt10b signaling regulates replication stress–induced chromosome missegregation and breakage, and hence is a determinant for broad genome instability in cancer cells.**

## Introduction

Wnt signaling pathways play central and diverse roles in development and tissue homeostasis (Nusse & Clevers, 2017; Albrecht et al, 2021). The best investigated example is the Wnt/β-catenin pathway that can be activated for instance by the Wnt3a ligand leading to stabilization and activation of the transcriptional regulator β-catenin. This, in turn, drives the expression of a multitude of target genes contributing to the various cellular functions of Wnt signaling (Cadigan & Waterman, 2012; Albrecht et al, 2021). Importantly, Wnt/β-catenin signaling is frequently up-regulated in human cancer, particularly in colorectal cancer, where mutations in adenomatous polyposis coli (APC) cause hyperstabilization of β-catenin (Schneikert & Behrens, 2007; Zhan et al, 2017). Consequently, Wnt target genes including genes regulating the cell cycle are aberrantly expressed, thereby contributing to tumorigenesis and tumor progression (He et al, 1998; Tetsu & Mccormick, 1999). Wnt signaling can also stabilize proteins other than β-catenin. This is referred to as the "Wnt Stabilization Of Proteins (Wnt/STOP)" pathway, which was shown to be involved in the regulation of cell size and mitosis (Taelman et al, 2010; Acebron et al, 2014; Huang et al, 2015; Stolz et al, 2015; Hinze et al, 2019). In fact, inhibition of β-catenin–independent Wnt signaling causes chromosome missegregation in mitosis leading to aneuploidy in human somatic cells and in pluripotent stem cells (Stolz et al, 2015; Augustin et al, 2017; Lin et al, 2021; Jaime-Soguero et al, 2024).

Perpetual mitotic chromosome missegregation promotes the evolvement of aneuploidy, which represents the basis for whole chromosome instability (W-CIN), a major form of genome instability and a hallmark of human cancer (Hanahan & Weinberg, 2011; Bakhoum & Landau, 2017; Sansregret et al, 2018; Lukow & Sheltzer, 2022; Chen et al, 2025). W-CIN originates during mitosis and can be driven by various abnormalities leading to chromosome missegregation (Bastians, 2015; Devillers et al, 2024). Specifically, an abnormal increase in growth rates of microtubules within the mitotic spindle, which impairs proper positioning of the spindle, has been recently identified as an important mechanism leading to W-CIN in human cancer cells (Ertych et al, 2014, 2016; Tamura et al, 2020; Schmidt et al, 2021; Paim et al, 2024). In fact, chromosomally unstable cancer cells that are characterized by high levels of chromosome missegregation typically display increased spindle microtubule growth rates as a driver for aneuploidy and thus for W-CIN (Ertych et al, 2014; Tamura et al, 2020; Bohly et al, 2022).

In contrast to W-CIN, structural CIN (S-CIN) causes structural chromosome aberrations including DNA amplifications, deletions, and translocations (Siri et al, 2021). S-CIN is often associated with chromosomal breaks that are driven by defects in DNA repair or by slowed or stalled DNA replication, a condition known as DNA replication stress (Zeman & Cimprich, 2014; Siri et al, 2021). Overall,

---

[1]Department of Molecular Oncology, Section for Cellular Oncology, Göttingen Center for Molecular Biosciences (GZMB), University Medical Center Göttingen (UMG), Göttingen, Germany   [2]Centre for Organismal Studies (COS), Heidelberg University, Heidelberg, Germany   [3]Karlsruhe Institute of Technology (KIT), Institute of Biological and Chemical Systems-Functional Molecular Systems (IBCS-FMS), Eggenstein-Leopoldshafen, Germany   [4]German Cancer Research Center (DKFZ), Division of Signaling and Functional Genomics, and Medical Faculty Heidelberg, Institute of Human Genetics, Department of Molecular Human Genetics, Heidelberg, Germany   [5]IKERBASQUE, Basque Foundation of Science, Bilbao, Spain

Correspondence: holger.bastians@uni-goettingen.de

CIN represents a major form of genome instability and is a hallmark of human cancer (Hanahan & Weinberg, 2011; Sansregret et al, 2018). Consequently, CIN promotes tumor evolution by generating genomic heterogeneity in cancer cells, thereby supporting aggressive growth phenotypes, metastasis, and therapy resistance (Bakhoum & Landau, 2017; Sansregret et al, 2018; Lukow & Sheltzer, 2022; Chen et al, 2025).

Interestingly, S-CIN and W-CIN are typically detected concomitantly in chromosomally unstable cancer cells suggesting functional links between the two forms of CIN (Janssen et al, 2011; Burrell et al, 2013; Tijhuis et al, 2019). Indeed, several studies showed that replication stress not only triggers S-CIN, but also affects mitosis leading to chromosome missegregation and this can be mediated by increased microtubule dynamics (Burrell et al, 2013; Bohly et al, 2019, 2022; Wilhelm et al, 2019; Dwivedi et al, 2023). In addition, we recently reported that in pluripotent stem cells, loss of Wnt signaling can impact DNA replication, thereby supporting the generation of mitotic errors (Jaime-Soguero et al, 2024). However, it remains unclear what mechanisms and pathways contribute to abnormal microtubule dynamics as a trigger for chromosome missegregation in response to replication stress. Here, we show that in human cancer cells Wnt10b signaling is required upon replication stress to ensure normal microtubule dynamics from the S phase until mitosis and to prevent chromosomal breaks and mitotic errors without affecting DNA replication dynamics per se. Hence, we propose that Wnt10b signaling acts in cancer cells during S phase–associated replication stress as a rescue pathway to limit genome instability.

# Results

### Loss of Wnt10b signaling causes mitotic errors

We have previously shown that loss of Wnt10b signaling causes mitotic errors in human somatic cells (Fig 1A). Of note, inhibition of Wnt(10b) signaling in chromosomally stable colorectal cancer cells (HCT116), either at the canonical (co)receptor level by DKK1 treatment, by knockout of the Wnt secretion factor *EVI/WNTLESS* (Augustin et al, 2017) or of the *WNT10B* ligand (Fig S1A) triggered increased microtubule growth rates in mitotic cells as determined by live-cell microscopy tracking of individual microtubule plus tips within mitotic spindles (Fig 1B and C, Video 1). Increased mitotic microtubule growth rates, in turn, were associated with increased chromosome missegregation during mitosis as detected by the enhanced occurrence of lagging chromosomes during anaphase of mitosis (Fig 1D and E). The causal relationship between increased microtubule growth rates and chromosome missegregation was validated by direct suppression of chromosome missegregation after restoration of proper microtubule growth rates using sub-nanomolar concentrations of the microtubule-binding drug Taxol as shown in previous studies (Fig 1C and E) (Ertych et al, 2014; Stolz et al, 2015). In all conditions of inhibited Wnt signaling, the mitotic errors were suppressed upon GSK3 kinase inhibition, indicating that Wnt-GSK3 signaling is involved in mitotic regulation, which is in agreement with our

previous findings (Fig 1C and E) (Lin et al, 2021; Jaime-Soguero et al, 2024). Importantly, only treatment with purified Wnt10b, but not with Wnt3a protein, rescued both abnormally increased mitotic microtubule growth rates and chromosome missegregation in HCT116-*WNT10B* and HCT116-*EVI/WNTLESS* knockout cells (Fig 1C and E). It is of note that Wnt10b was largely unable to activate Wnt/β-catenin signaling as detected by Wnt reporter assays in three different cell lines when compared to Wnt3a (Fig S1B) supporting the notion that β-catenin–independent Wnt10b signaling is required for faithful chromosome segregation (Stolz et al, 2015; Lin et al, 2021).

### Wnt10b signaling is required during the S phase to ensure faithful mitotic chromosome segregation

Because Wnt10b signaling is required for proper mitosis, we expected it to act immediately before or during mitosis to ensure normal mitotic microtubule dynamics and faithful chromosome segregation. To test this, we treated cell cycle–synchronized HCT116 cells with DKK1 to induce mitotic errors or synchronized HCT116-*EVI/WNTLESS* knockout cells with purified Wnt10b to rescue mitotic errors at specified phases of the cell cycle (Figs 2A and S2A). Surprisingly, DKK1-mediated inhibition of Wnt signaling for only 2 h during the S phase, but not during the G2 phase or immediately before or during mitosis (G2/M) induced GSK3-dependent abnormal microtubule growth rates and chromosome missegregation (Fig 2B and C). Vice versa, treatment of HCT116-*EVI/WNTLESS* knockout cells with purified Wnt10b ligand rescued mitotic errors only when applied for 2 h during the S phase, but not during later stages of the cell cycle, whereas treatment with Wnt3a from the S phase until mitosis had no effect (Fig 2D and E). Also, abnormally increased microtubule dynamics in HCT116-*EVI/WNTLESS* knockout cells were rescued by treatment with the GSK3 inhibitor during the S phase, whereas wash-off of the inhibitor just before entry into mitosis had no effect (Fig S2B). GSK3 inhibition shortly before cells entered mitosis had no effect on mitotic microtubule growth rates (Fig S2B). Together, our results indicate that GSK3-dependent Wnt10b signaling is required specifically during the S phase to ensure faithful chromosome segregation. To extend these findings, we used chromosomally unstable (CIN+) colorectal cancer cells, which are characterized by inherently high rates of chromosome missegregation that are caused by increased microtubule growth rates (Ertych et al, 2014; Tamura et al, 2020; Bohly et al, 2022). Also, in these CIN+ cancer cells treatment with Wnt10b, but not Wnt3a, specifically during the S phase efficiently rescued abnormal microtubule growth rates and chromosome missegregation in mitosis (Fig 2F and G) demonstrating that Wnt10b signaling is important during the S phase to suppress mitotic errors in human cancer cells.

### Inhibition of Wnt signaling increases microtubule growth rates from the S phase into mitosis

Because Wnt10b signaling is required during the S phase but regulates mitosis several hours later, we wondered whether inhibition of Wnt signaling impacts microtubule behavior already before cells enter mitosis. Thus, we determined interphase

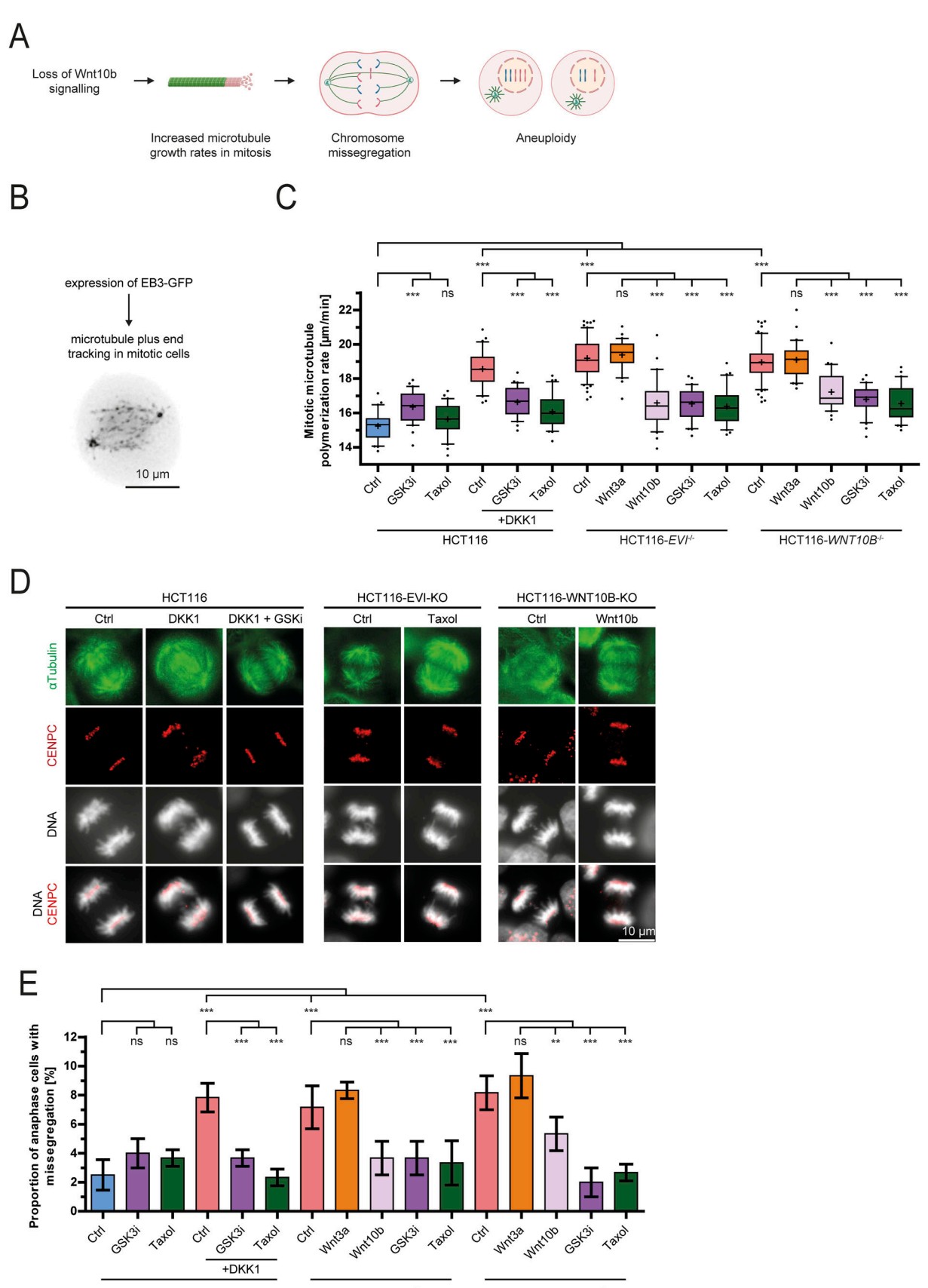

microtubule growth rates in synchronized cells with or without Wnt inhibition (Fig 3A and B). Interestingly, DKK1 treatment or loss of *EVI/WNTLESS* resulted in increased microtubule growth rates in the S phase, and this abnormality was maintained until mitosis (Fig 3C). The same findings were obtained using synchronized CIN+ SW480 and SW620 cells that exhibited inherently high microtubule polymerization rates from the S phase until mitosis (Fig 3D). Importantly, treatment of synchronized Wnt-inhibited or CIN+ cancer cells with low doses of Taxol just before mitosis (Fig 3E) was sufficient to restore normal microtubule growth rates and suppressed chromosome missegregation (Fig 3F and G), whereas treatment with DKK1 had no additional effect on microtubule dynamics and chromosome missegregation in CIN+ cancer cells (Fig S3A and B). Thus, abnormal microtubule growth rates originate from the S phase, and are maintained until mitosis where they are ultimately causing chromosome missegregation.

### Loss of Wnt signaling does not cause replication stress, but leads to chromosome missegregation

Because Wnt inhibition acts specifically during the S phase to regulate mitosis, we investigated whether Wnt10b signaling might be linked to DNA replication, which is the main process during the S phase of the cell cycle. In fact, it is well established that DNA replication stress is a frequent condition in chromosomally unstable cancer cells and associated with broad genome instability (Zeman & Cimprich, 2014; Gaillard et al, 2015). Moreover, recent studies have highlighted that replication stress also impacts mitosis and causes abnormally increased microtubule dynamics, thereby explaining the concomitant presence of replication stress and high chromosome missegregation rates in chromosomally unstable cancer cells (Burrell et al, 2013; Liu et al, 2014; Fragkos & Naim, 2017; Bohly et al, 2019, 2022; Wilhelm et al, 2020). In agreement with this, we found that induction of replication stress in chromosomally stable HCT116 cells by treatment with low concentrations of aphidicolin, a specific inhibitor of DNA polymerases and well-established inducer of replication stress (Ikegami et al, 1978; Minocherhomji et al, 2015; Bohly et al, 2019, 2022), increased mitotic microtubule growth rates and chromosome missegregation (Fig 4A and B). Vice versa, increased microtubule growth rates and chromosome missegregation endogenously present in CIN+ cancer cells were suppressed upon deoxynucleoside supplementation, an established mean to alleviate replication stress (Burrell et al, 2013; Wilhelm et al, 2014; Bohly et al, 2019) (Fig 4C and D). Although deoxynucleoside supplementation might affect microtubule dynamics and mitosis also by other yet not investigated mechanisms, for example, by

affecting purinergic signaling (Huang et al, 2021), these results strongly suggest that DNA replication stress present in chromosomally unstable cancer cells causes abnormal mitotic microtubule dynamics and chromosome missegregation as demonstrated previously (Bohly et al, 2019, 2022). Thus, replication stress during the S phase mimics Wnt inhibition with respect to the induction of mitotic errors raising the question whether Wnt10b inhibition causes replication stress. To address this, we employed DNA combing analysis as the gold standard method to measure the progression of individual replication forks during DNA replication (Fig 4E) (Moore et al, 2022). As expected, treatment of cells with low concentrations of aphidicolin caused mild replication stress as indicated by reduced replication fork velocities (Fig 4F). Importantly, neither DKK1 treatment of HCT116 cells nor the loss of *EVI/WNTLESS* (HCT116-*EVI* knockout cells) grossly affected replication fork speed (Figs 4F and S4A). Accordingly, no reduced inter-origin distances were found upon Wnt inhibition (Fig S4A), which is known to be a direct consequence of replication stress (Ibarra et al, 2008; Moiseeva & Bakkenist, 2019; Bohly et al, 2022). We also performed EdU-FACS analysis to quantify S phase–associated nucleotide incorporation as an alternative approach to assess replication stress (Macheret & Halazonetis, 2019). Although these assays clearly demonstrated reduced nucleotide incorporation upon aphidicolin treatment as expected, DKK1 treatment had no effect (Fig S4B) further supporting that inhibition of Wnt signaling does not cause global replication stress per se. Furthermore, in agreement with previous studies (Burrell et al, 2013; Bohly et al, 2019, 2022), we detected reduced fork progression rates and thus endogenous mild replication stress in CIN+ cancer cells, but treatment with Wnt10b or Wnt3a did not grossly affect replication fork progression dynamics in these cells (Fig 4F) or in RPE1-hTERT cells treated with aphidicolin (Fig S4C), indicating that activation of Wnt signaling does not alleviate replication stress. Severe replication stress can be associated with cell cycle delay or arrest in the S phase, which is known to be mediated by ATR-Chk1 kinase–dependent cell cycle checkpoint signaling (Bartkova et al, 2005; Gorgoulis et al, 2005; Zeman & Cimprich, 2014). Indeed, we detected phosphorylation and hence activation of ATR and Chk1 kinases upon treatment with increasing concentrations of aphidicolin, but no checkpoint activation was seen upon DKK1-mediated Wnt inhibition (Fig S4D). In the same line, we found no alterations in cell cycle stage distribution neither upon DKK1 nor upon GSK3 inhibitor treatment (Fig S5A). Also, FACS analysis of synchronized cell populations showed timely cell cycle progression from G1/S until entry into mitosis of cells treated with DKK1 or upon *EVI/WNTLESS* knockout (Fig S5B and C). Together, we

---

**Figure 1. Wnt10b signaling is required for normal microtubule dynamics and faithful chromosome segregation during mitosis.**
**(A)** Model depicting the relationship between the loss of Wnt signaling, increased mitotic microtubule plus–end growth rates, chromosome missegregation, and the induction of aneuploidy. **(B)** Experimental outline for the measurement of microtubule plus–end growth rates. **(C)** Measurements of mitotic microtubule growth rates in HCT116 cells with or without knockout of *EVI/WNTLESS* or *WNT10B* and additional treatment as indicated. Cells were treated with DMSO (Ctrl), 600 ng/ml DKK1, 0.6 $\mu$M GSK3 inhibitor (CHIR-99021), 0.2 nM Taxol, or 400 ng/ml of recombinant Wnt ligands. Measurements are based on analysis of 25 microtubules/cell (n ≥ 30 cells from three independent experiments, two-tailed *t* test). **(D)** Example images of chromosome segregation during anaphase in HCT116, HCT116-*EVI*$^{-/-}$, and HCT116-*WNT10B*$^{-/-}$ cells treated as indicated. Fixed cells were stained for spindle microtubules ($\alpha$-tubulin), kinetochores (Cenp-C), and chromosomes (DNA) to detect lagging chromatids during anaphase of mitosis; scale bar, 10 $\mu$m. **(E)** Quantification of the proportion of cells exhibiting chromosome missegregation upon Wnt inhibition. **(C)** Cells were treated as described in (C), and anaphase cells with lagging chromosomes were quantified. Graphs show mean values ± SD (n = 300 cells from three independent experiments, two-tailed *t* test).

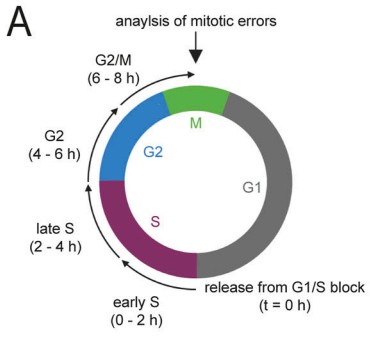

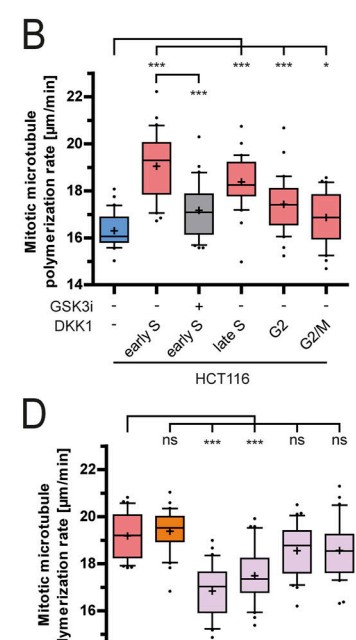

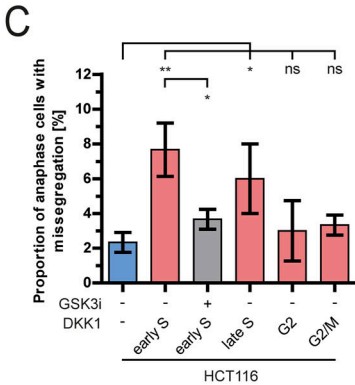

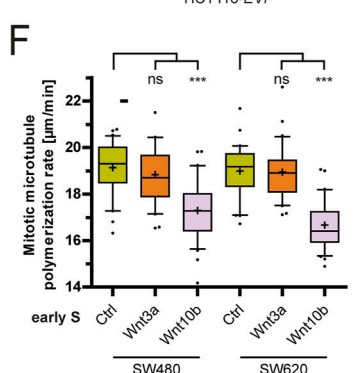

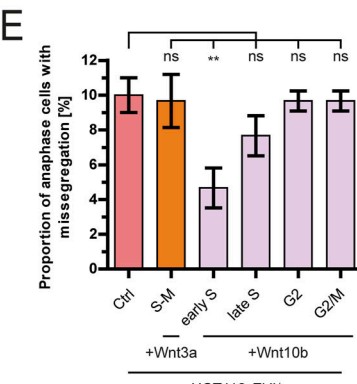

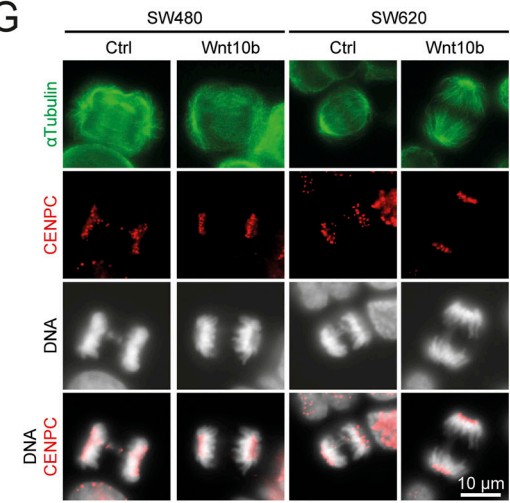

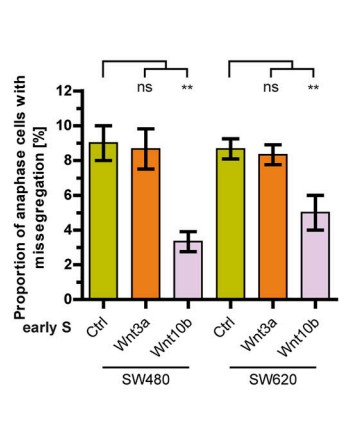

**Figure 2. Wnt10b acts during the S phase to promote faithful mitosis.**
**(A)** Schematic outlining cell cycle–specific Wnt modulation using cell cycle–synchronized cells followed by analysis of mitotic outcomes. **(B)** Measurements of microtubule growth rates in HCT116 cells after cell cycle phase–specific treatment with 600 ng/ml DKK1 in the absence or presence of 0.6 $\mu$M GSK3 kinase inhibitor (25 microtubules/cell, n = 30 cells from three independent experiments, two-tailed $t$ test). **(C)** Quantification of anaphase cells showing chromosome missegregation upon treatments as described in (B). Graph shows mean values ± SD (n = 300 cells from three independent experiments, two-tailed $t$ test). **(D)** Measurements of microtubule growth rates in HCT116-*EVI/WNTLESS* knockout cells after cell cycle–specific treatment with 400 ng/ml of recombinant Wnt ligands (25 microtubules/cell, n = 30 cells from three independent experiments, two-tailed $t$ test). **(E)** Quantification of HCT116-*EVI/WNTLESS* knockout cells showing chromosome missegregation after treatments as described in (D) (mean ± SD, n = 300 cells from three independent experiments, two-tailed $t$ test). **(F)** Measurements of microtubule growth rates in chromosomally unstable (CIN+) SW480 and SW620 colorectal cancer cells with or without treatment with 400 ng/ml Wnt ligands during the early S phase for 2 h (25 microtubules/cell, n = 30 cells from three independent experiments, two-tailed $t$ test). **(G)** Left: example images of CIN+ cells in anaphase with and without Wnt10b treatment stained for spindle microtubules ($\alpha$-tubulin), kinetochores (Cenp-C), and chromosomes (DNA); scale bar, 10 $\mu$m. Right: quantification of CIN+ cancer cells displaying chromosome missegregation with or without treatment with Wnt10b (mean ± SD, n = 300 cells from three independent experiments, two-tailed $t$ test).

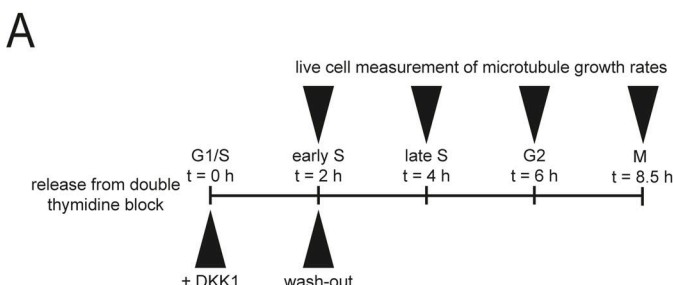

A

live cell measurement of microtubule growth rates

release from double thymidine block

G1/S t = 0 h early S t = 2 h late S t = 4 h G2 t = 6 h M t = 8.5 h

± DKK1 wash-out

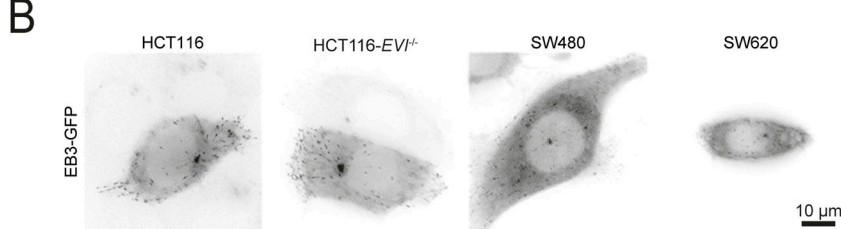

B

EB3-GFP

HCT116 HCT116-*EVI*⁻/⁻ SW480 SW620

10 μm

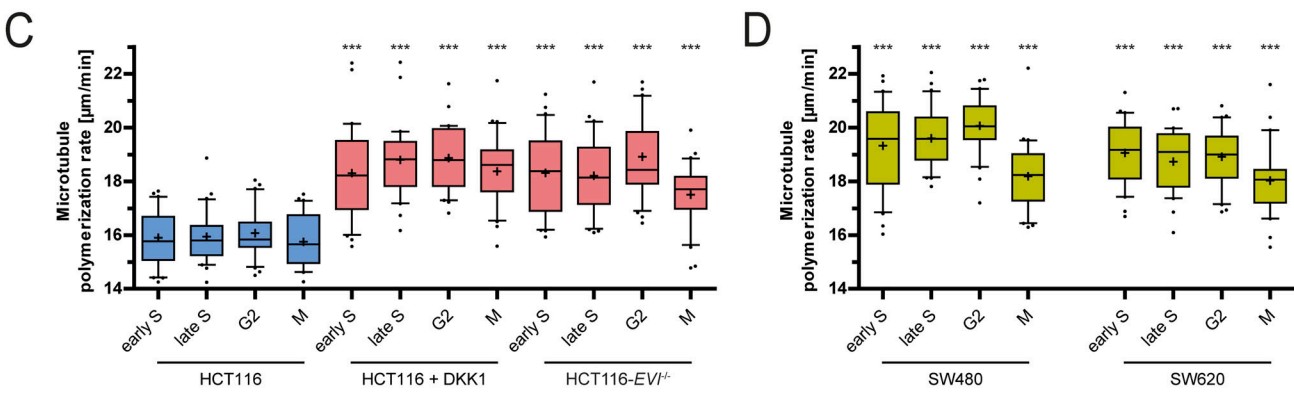

C

D

Microtubule polymerization rate [μm/min]

E

release from double thymidine block

± DKK1 wash-out ± Taxol mitotic errors?

G1/S t = 0 h early S t = 2 h late S t = 4 h G2 t = 6 h M t = 8.5 h

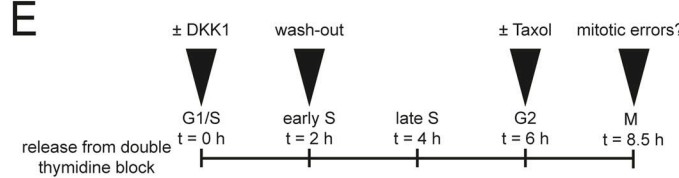

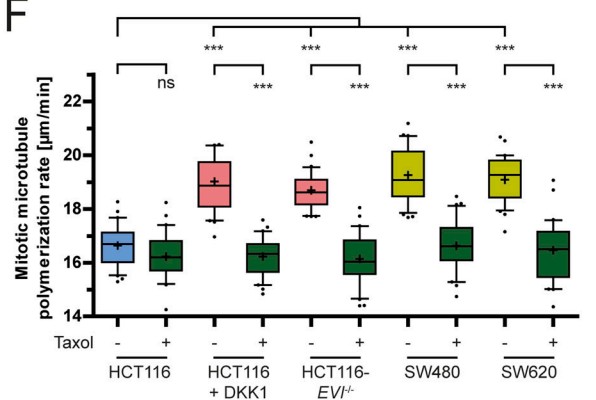

F

Mitotic microtubule polymerization rate [μm/min]

Taxol − + − + − + − + − +

HCT116 HCT116 + DKK1 HCT116-*EVI*⁻/⁻ SW480 SW620

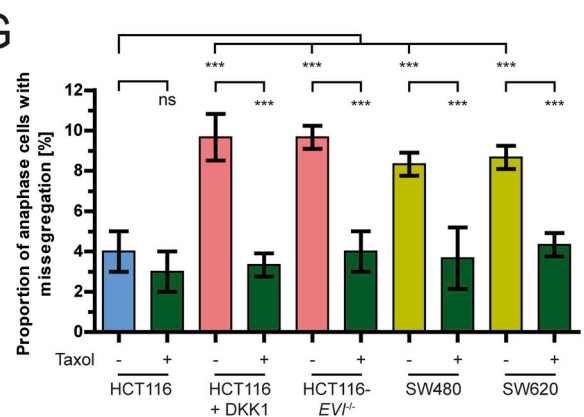

G

Proportion of anaphase cells with missegregation [%]

Taxol − + − + − + − + − +

HCT116 HCT116 + DKK1 HCT116-*EVI*⁻/⁻ SW480 SW620

conclude that inhibition of Wnt10b signaling leads to mitotic chromosome missegregation, but Wnt inhibition impacts neither DNA replication dynamics nor cell cycle progression in human cancer cells.

### Wnt10b signaling acts downstream of replication stress to ensure faithful mitosis

Because Wnt inhibition does not impact replication dynamics per se, we hypothesized that Wnt10b signaling might act downstream of replication stress to ensure proper mitosis. This hypothesis was driven by the Wnt10b-mediated rescue of mitotic errors in CIN+ cells (Fig 2F and G), where mitotic errors are triggered by endogenous replication stress (Fig 4C, D, and F). To further address this, we induced replication stress during the early S phase by pulse treatment with aphidicolin or hydroxyurea, which inhibits DNA polymerase and depletes cellular nucleotide pools, respectively (Fig 5A) (Bianchi et al, 1986). Both means of replication stress induction resulted in increased microtubule growth rates and chromosome missegregation in the subsequent mitosis, both of which were suppressed by co-treatment with Wnt10b, but not with Wnt3a (Fig 5B and C). In addition, we also determined microtubule growth rates during the S phase immediately upon induction of replication stress. In fact, replication stress induced increased microtubule growth rates already in the S phase where they were suppressed by Wnt10b treatment (Fig 5D). Very similarly, CIN+ cancer cells suffering from endogenous replication stress (Fig 4F) also exhibited increased microtubule dynamics in the S phase, which was rescued by Wnt10b addition (Fig 5D). It is of note that induction of replication stress by aphidicolin or alleviation of endogenous replication stress in CIN+ cancer cells by deoxynucleoside treatment did not influence canonical Wnt signaling as determined by the expression of the well-established $\beta$-catenin/TCF target gene *AXIN2* (Fig S6A and B). Together, these data indicate that Wnt10b signaling acts during the S phase to suppress mitotic errors induced by DNA replication stress in human cancer cells.

### Wnt10b signaling suppresses replication stress–induced chromosome breaks

In addition to causing mitotic errors, replication stress is known to give rise to chromosomal breaks, which form the basis of structural chromosome instability in cancer (Branzei & Foiani, 2010; Zeman & Cimprich, 2014). Because Wnt10b regulates mitotic errors in response to replication stress, we wondered whether Wnt10b signaling also affects the downstream generation of chromosomal breaks. To address this, we induced replication stress with aphidicolin or hydroxyurea treatment during the early S phase, either in the presence or in the absence of Wnt10b or Wnt3a (Fig 6A). As expected, we observed increased chromosomal breakage upon replication stress as determined by chromosome spread analysis (Fig 6B). Intriguingly, treatment with Wnt10b, but not with Wnt3a, significantly suppressed replication stress–induced chromosomal breaks (Fig 6C), indicating that Wnt10b indeed modulates the generation of chromosomal breaks upon replication stress. Similarly, increased levels of chromosomal breaks were also detectable in CIN+ cancer cells, which were suppressed upon alleviation of replication stress upon deoxynucleoside supplementation, indicating that these chromosomal breaks resulted from endogenous replication stress present in these cancer cells (Fig 6D). Importantly, the replication stress–induced chromosomal breaks in these CIN+ cancer cells were also efficiently suppressed upon Wnt10b, but not Wnt3a treatment (Fig 6D). Further supporting the role of Wnt10b signaling in suppressing chromosomal breaks downstream of replication stress, we found that inhibition of Wnt signaling either upon DKK1 treatment or in *EVI/WNTLESS* knockout cells induced Wnt10b-dependent chromosomal breaks as seen after replication stress (Fig 6C). Taken together, our results indicate that Wnt10b signaling functions as a rescue pathway downstream of replication stress, preventing both mitotic chromosome missegregation and chromosome breaks, two well-established consequences of DNA replication stress and hallmarks of genome instability in human cancer.

## Discussion

Our work revealed a yet unrecognized and unexpected function of Wnt10b signaling to protect cells from mitotic errors and chromosomal breaks in response to DNA replication stress, which is a major source for chromosomal instability (CIN) in human cancer (Zeman & Cimprich, 2014; Igarashi et al, 2024). Thus, Wnt10b signaling might act as a suppressor of CIN that is a well-established driving force for the generation of high genetic heterogeneity and variability in cancer supporting tumor evolution toward aggressive growth phenotypes, metastasis, and therapy resistance (Sansregret et al, 2018; Chen et al, 2025).

Replication stress (RS) is a highly cancer-relevant condition that is characterized by slowed or stalled DNA replication during the S phase of the cell cycle. It can be triggered by various cellular conditions including nucleotide shortage, the presence of abnormal chromosome structures, or activation of oncogenes such as *MYC*, *RAS*, or *CYCLIN-E* and contributes to the induction of structural chromosome aberrations that are frequently seen in cancer (Zeman & Cimprich, 2014; Igarashi et al, 2024). In addition, recent work has uncovered that RS also affects mitosis by

**Figure 3. Wnt inhibition causes increased microtubule growth rates from the S phase until mitosis.**
**(A)** Experimental setup for the cell cycle–specific analysis of microtubule growth rates. **(B)** Example images of live cells synchronized in the S phase and expressing GFP-EB3 to measure interphase microtubule growth rates; scale bar, 10 $\mu$m. **(C)** Cell cycle stage–dependent measurements of microtubule growth rates after inhibition of Wnt signaling in HCT116 upon DKK1 treatment or in HCT116-*EVI/WNTLESS* knockout cells (20 microtubules/cell, n = 30 cells from three independent experiments). **(D)** Cell cycle stage–dependent measurements of microtubule growth rates in chromosomally unstable colorectal cancer cells (20 microtubules/cell, n = 30 cells from three independent experiments). **(E)** Experimental setup for the analysis of mitotic microtubule assembly rates upon 0.2 nM Taxol treatment. **(F)** Measurements of mitotic microtubule growth rates in the indicated cells with or without Taxol treatment at G2/M (20 microtubules/cell, n = 30 cells from three independent experiments). **(F, G)** Quantification of cells showing chromosome missegregation after treatment as in (F) (mean ± SD, n = 300 cells from three independent experiments, two-tailed *t* test).

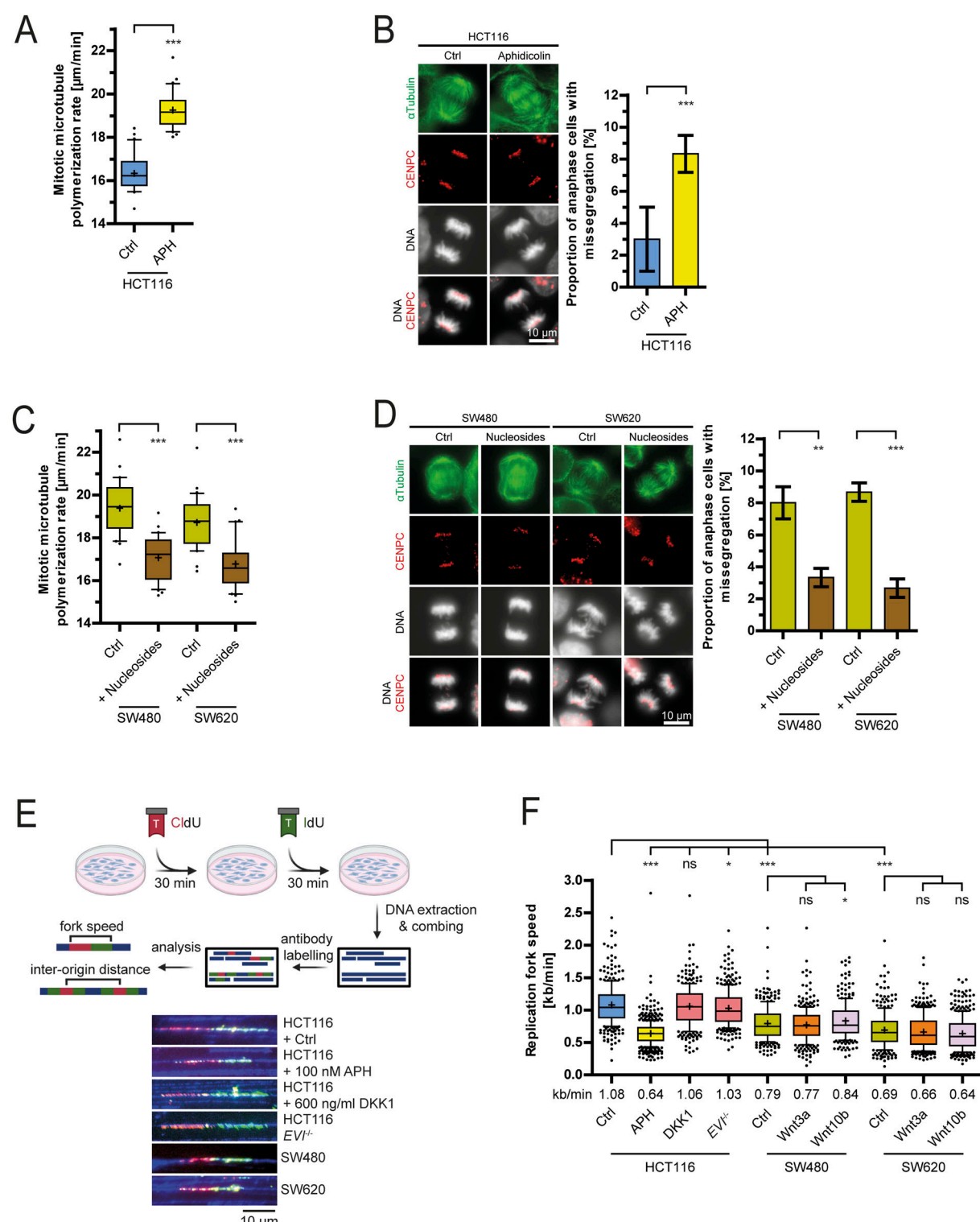

**Figure 4. Replication stress does not link Wnt signaling and chromosome missegregation.**
**(A)** Measurement of microtubule polymerization rates in HCT116 cells after induction of replication stress. Cells were treated with 100 nM aphidicolin (APH) for 16 h before live-cell measurements of microtubule growth rates (25 microtubules/cell, n = 30 cells from three independent experiments, two-tailed *t* test). **(B)** Left: example images of HCT116 cells in anaphase treated with and without aphidicolin and stained for spindle microtubules (α-tubulin), kinetochores (Cenp-C), and chromosomes (DNA); scale bar, 10 μm. Right: quantification of anaphase cells showing chromosome missegregation upon replication stress. HCT116 cells were treated with 100 nM aphidicolin for 16 h, and chromosome missegregation was detected in anaphase cells (mean ± SD, n = 300 cells from three independent experiments, two-tailed *t* test). **(C)** Measurement of mitotic microtubule growth rates in CIN+ cancer cells upon alleviation of replication stress by deoxynucleoside supplementation (25 microtubules/cell, n = 30 cells from three independent experiments, two-tailed *t* test). **(D)** Left: example images of CIN+ cells in anaphase with and without deoxynucleoside supplementation and stained for spindle microtubules (α-tubulin), kinetochores (Cenp-C), and chromosomes (DNA); scale bar, 10 μm. **(C)** Right:

deregulating mitotic microtubule dynamics and centrosome separation leading to chromosome missegregation, thereby driving the evolvement of aneuploidy (Burrell et al, 2013; Bohly et al, 2019, 2022; Wilhelm et al, 2019; Jaime-Soguero et al, 2024). With this, RS sits at the helm of CIN in cancer cells.

Intriguingly, our work demonstrates that activation of Wnt10b signaling in cancer cells suffering from RS is sufficient to suppress both the generation of chromosomal breaks and the generation of mitotic errors. However, Wnt10b activation does not alleviate global RS indicating that Wnt signaling acts downstream of RS to suppress the generation of chromosomal breaks and to ensure faithful mitosis. An attractive hypothesis is that Wnt10b signaling might be involved in maintaining the integrity of replication forks. In fact, stalled replication forks are known to require stabilization, remodeling, and repair in order to allow efficient restart to achieve complete DNA replication (Qiu et al, 2021). Wnt10b activation might foster such integrity mechanisms through activating and/or stabilizing proteins that contribute to improved fork repair leading to reduced chromosomal breaks. In this context, it is possible that Wnt10b might also act in response to other stress conditions such as general DNA damage to improve DNA repair, but this remains to be tested.

Notably, Wnt10b activation suppresses increased microtubule dynamics, which is induced upon RS in the S phase and maintained until mitosis where this cellular abnormality causes chromosome missegregation. Whether alterations in microtubule dynamics during S-phase are determinants for the generation of chromosomal breaks is unknown, but it is tempting to speculate that Wnt10b-regulated microtubule dynamics might affect fork maintenance and/or repair. Interestingly, recent reports have indicated that dynamic microtubules are involved in nuclear DNA repair through regulation of chromatin mobility involving the microtubule–LINC (linker of nucleoskeleton and cytoskeleton) complex (Lottersberger et al, 2015; Lawrence et al, 2016). Such microtubule-involving repair mechanisms might be activated upon RS and positively regulated by Wnt10b signaling. In this case, one could speculate that cancer cells might exhibit dysfunctional Wnt10b signaling leading to impaired fork maintenance and repair resulting in elevated levels of chromosomal breaks and CIN. To further investigate this, it would be useful to identify target proteins that are involved in such functions and regulated specifically by Wnt10b, but not by other Wnts.

In this study, we showed that Wnt10b signaling acts downstream of RS, but does not impact DNA replication dynamics in cancer cells per se. This stands in contrast to our recent findings in pluripotent stem cells where Wnt3a signaling affects replication directly by suppressing replication origin firing in the S phase (Jaime-Soguero et al, 2024). The difference between stem cells and somatic cells might be explained by the fact that stem cells are characterized by a much longer duration of DNA replication, and thus, they might be more prone to RS. In fact, it has been shown that pluripotent stem cells have high basal levels of RS and unique replication profiles (Kafer et al, 2022 Preprint; Kurashima et al, 2024). So, it is possible

that Wnt3a signaling in stem cells is fulfilling similar roles as Wnt10b signaling in somatic cells. Further analysis on proteins that are regulated by Wnt3a and Wnt10b during the S phase in stem cells and somatic cells, respectively, is required to define the overlapping or nonoverlapping pathways involved in the S phase and mitotic regulation in both cell systems.

Another intriguing finding of this study is that specifically Wnt10b and not Wnt3a acts during the S phase of the cell cycle to limit chromosomal breaks and mitotic errors in the presence of RS, although both Wnt ligands were suggested to signal through a cascade involving Frizzled receptors, LRP co-receptors, GSK3$\beta$ kinase, and the well-characterized destruction complex (Wend et al, 2012). Our previous work supports an involvement of the "canonical pathway components" and showed that LRP5/6, AXIN1, and GSK3$\beta$ kinase, but not $\beta$-catenin, are involved in the regulation of mitotic chromosome segregation (Stolz et al, 2015; Lin et al, 2021). Thus, we suggest that Wnt10b-dependent Wnt/STOP (Wnt-mediated stabilization of proteins) (Stolz & Bastians, 2015; Acebron & Niehrs, 2016), which employs the "classical" pathway, is required downstream of RS to regulate protein(s) other than $\beta$-catenin to limit breaks and mitotic errors. However, these Wnt10b targets are yet unknown and it is still unclear whether such targets are subject to Wnt-regulated phosphorylation and/or protein stabilization. Because it is known that different Wnts bind to different Frizzled receptors (FZDs), one can speculate that the different WNT-FZD complexes recruit different signaling complexes to the membrane that might involve different subpools of kinases directing them to individual substrates, thereby explaining why Wnt10b targets a different set of proteins compared with Wnt3a and other Wnts. Clearly, further detailed work is needed to solve these important and highly relevant questions.

Importantly, our work highlights Wnt10b signaling as an extracellular signaling pathway acting downstream of DNA RS to limit chromosomal breaks and mitotic errors. Intriguingly, we recently reported similar roles of other extracellular signaling pathways such as BMP and FGF signaling (Jaime-Soguero et al, 2024). Also, BMP signaling was recently shown to regulate RS during heart regeneration in zebrafish (Vasudevarao et al, 2025), together suggesting key roles not only of Wnt signaling, but also of extracellular cues per se in the regulation of genome stability.

# Materials and Methods

### Cell culture

HCT116 (RRID:CVCL_0291), SW480 (RRID:CVCL_054), SW620 (RRID:CVCL_0547), and HEK293T (RRID:CVCL_0063) cells were purchased from the ATCC. HCT116-*EVI*/*WNTLESS* knockout cells were described previously (Augustin et al, 2017; Lin et al, 2021). These cell lines were cultured in RPM1-1640 or DMEM (PAN Biotech) supplemented with 10% FBS (#AC-AB-0024; Anprotec) and 1% penicillin/streptomycin

---

quantification of anaphase cells showing chromosome missegregation upon treatment as in (C) (mean ± SD, n = 300 cells from three independent experiments, two-tailed *t* test). **(E)** Schematic setup for DNA combing to determine the replication fork speed. Representative examples of labeled unidirectional DNA fibers are shown. **(F)** Determination of replication fork progression rates using the indicated cells and treatments (>300 fibers per condition, two-tailed *t* test).

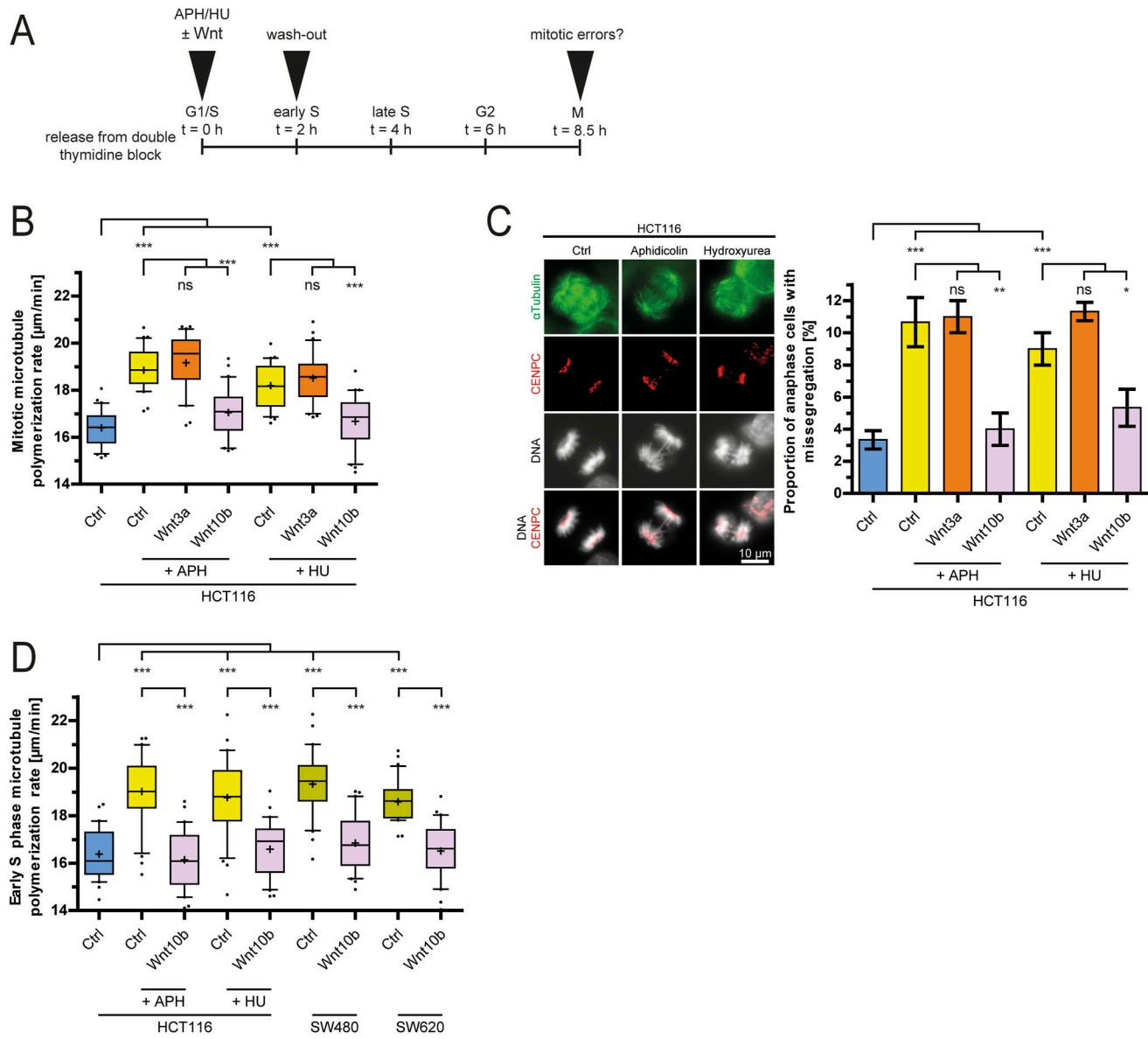

**Figure 5. Wnt10b signaling ensures faithful chromosome segregation after replication stress.**
**(A)** Experimental outline for the analysis of mitotic errors after replication stress and Wnt treatment. **(B)** Measurement of mitotic microtubule growth rates in HCT116 cells after treatment with 100 nM aphidicolin (APH) or 2 mM hydroxyurea (HU) in the presence or absence of 400 ng/ml Wnt ligands (25 microtubules/cell, n = 30 cells from three independent experiments, two-tailed $t$ test). **(C)** Quantification of anaphase cells showing chromosome missegregation after treatment as in (B) (mean ± SD, n = 300 cells from three independent experiments, two-tailed $t$ test). **(D)** Measurement of microtubule growth rates in the early S phase upon replication stress. HCT116, SW480, and SW620 cells were synchronized in the early S phase, and treated as indicated, and microtubule growth rates were analyzed during the S phase (25 microtubules/cell, n = 30 cells from three independent experiments, two-tailed $t$ test).

(#AC-AB-0024; Anprotec). RPE1-hTERT were kindly provided by Zuzana Storchova and cultured in DMEM/F12 (PAN Biotech) supplemented with 10% FBS (#AC-SM-0184; Anprotec), 1% penicillin/streptomycin (#AC-AB-0024; Anprotec), and 0.26% $NaHCO_3$ (PAN Biotech). Cells were maintained in a $CO_2$ incubator at 37°C with 5% $CO_2$.

## Generation of *WNT10B* knockout cells

HCT116-*WNT10B* knockout cells were generated as previously described (Ran et al, 2013; Voloshanenko et al, 2017). A short-guide RNA for CRISPR/Cas9 (3′-GGAAGAATGCGGCTCTGACA-5′) was designed using E-CRISP (http://www.e-crisp.org), purchased from Eurofins Inc., and cloned into a pSpCas9(BB)-2A-Puro (px459) plasmid, which was a gift from Feng Zhang (plasmid #48139; Addgene; http://n2t.net/addgene:48139; RRID:Addgene_48139) (Ran et al, 2013). HCT116 cells were transfected and selected in culture medium containing 2 μg/ml of puromycin. Pools of cells were first expanded, and a single-cell clone was generated and further analyzed.

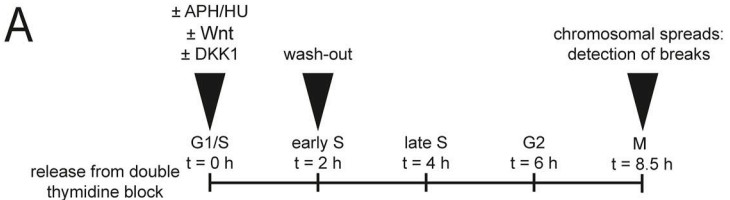

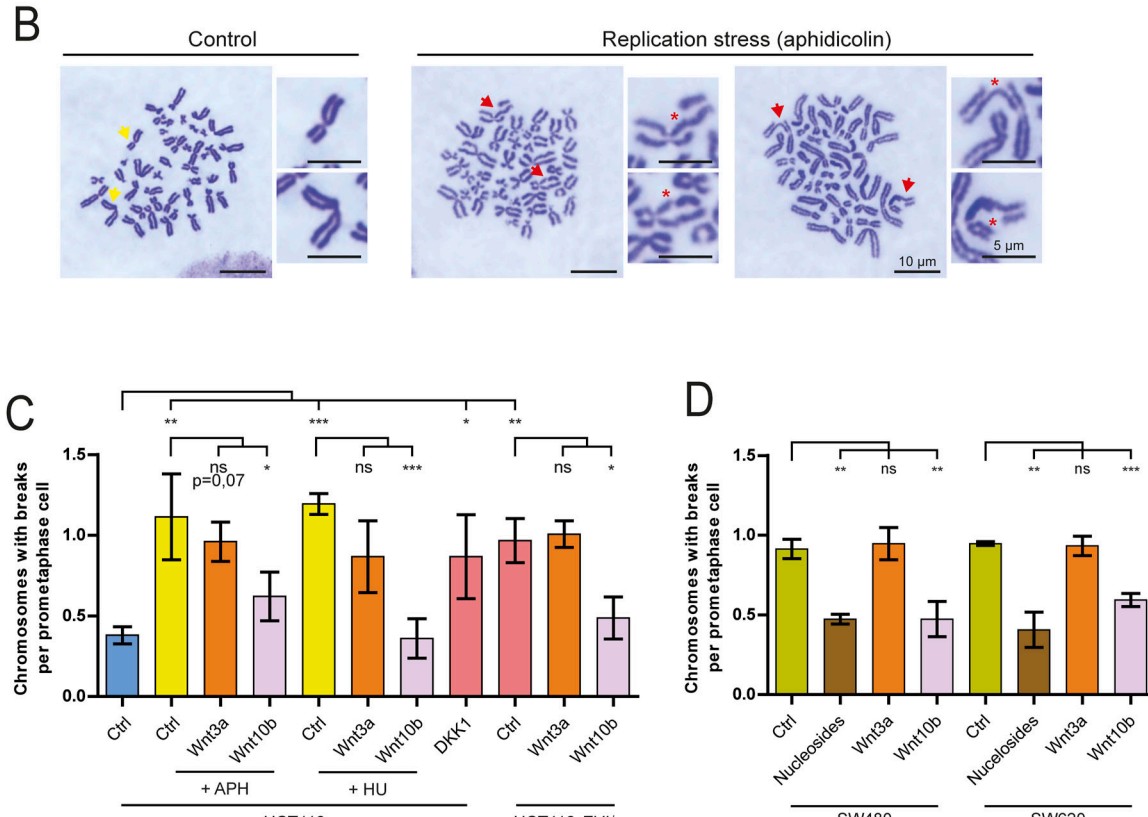

**Figure 6. Wnt10b suppresses replication stress–induced chromosomal breaks.**
**(A)** Setup for the analysis of chromosomal breaks. **(B)** Representative images of chromosome spreads with or without replication stress–induced chromosomal breaks. Scale bar, 5 μm and 10 μm as indicated. **(C)** Quantification of chromosome breaks after replication stress or Wnt inhibition. Mitotic chromosome spreads were analyzed from cells after induction of replication stress (100 nM APH or 2 mM HU) or after Wnt inhibition (DKK1, *EVI/WNTLESS* knockout) in the presence or absence of Wnt ligands. The graph shows the number of chromosomes with breaks per cell (mean ± SD, n = 150 chromosome spreads from three experiments; two-tailed *t* test). **(D)** Quantification of chromosome breaks in CIN+ cancer cells after treatment with Wnt ligands or upon deoxynucleoside supplementation. The graph shows the number of chromosomes with breaks per cell (mean ± SD, n = 150 chromosome spreads from three experiments, two-tailed *t* test).

## Cell treatments

Where indicated, cells were treated with 100 nM aphidicolin (sc-201535; Santa Cruz) or 2 mM hydroxyurea (H8627-1G; Merck-Millipore) to induce replication stress. Replication stress was alleviated by treatment with additional nucleosides for 48 h (30 μM 2′-deoxyadenosine monohydrate [#sc-216290; Santa Cruz], 30 μM 2′-deoxycytidine hydrochloride [#sc-220820; Santa Cruz], 30 μM thymidine [#sc-296542; Santa Cruz], and 30 μM 2′-deoxyguanosine monohydrate [#sc-238433; Santa Cruz]) as described previously (Wilhelm et al, 2014; Bohly et al, 2019). Cells were treated with 0.6 μM CHIR99021 (SML1046;

Sigma-Aldrich) to inhibit GSK3 kinase. Increased microtubule growth rates were suppressed by treatment with 0.2 nM Taxol (paclitaxel, T7191-1MG; Sigma-Aldrich) as shown previously (Ertych et al, 2014). As controls, cells were treated with $H_2O$ or dimethyl sulfoxide (D1418; Sigma-Aldrich). Cells were treated with 400 ng/ml of human recombinant Wnt3a (5036-WN/CF; R&D Systems) or human recombinant Wnt10b (7196-WN/CF; R&D Systems) to induce ligand-specific Wnt signaling, or with 600 ng/ml of human recombinant DKK1 (120-30-500; Pepro-Tech) to inhibit Wnt/LRP6 signaling. 0.1% BSA (8076.2; Carl Roth) in PBS was used as a control treatment for all recombinant proteins.

## Wnt reporter assay

Cells were seeded in a 96-well dish and were transfected with 20 ng TCF firefly luciferase (TOPFLASH) plasmid and 2 ng CMV Renilla luciferase plasmid using ScreenFect A according to the manufacturer's protocol (ScreenFect GmbH). After 24 h, cells were treated with purified Wnt ligands and incubated overnight. Cells were harvested in 1x passive lysis buffer (Promega) and processed according to the manufacturer's protocol. TOPFLASH luciferase values were normalized to control Renilla luciferase values. Mean values (±SD) were calculated from four independent experiments.

## qRT-PCR analysis

RNA was extracted and purified using the RNeasy Plus column kit (QIAGEN), according to the manufacturer's instructions. The cDNA was produced with iScript cDNA Synthesis Kit (Bio-Rad) using 300 ng mRNA. Quantitative real-time PCRs were set up from technical triplicates using SsoAdvanced Universal SYBR Green Supermix (Bio-Rad) on a QuantStudio 5 (Thermo Fisher Scientific). mRNA expression levels were normalized to *GAPDH*. The following primers were used:

*AXIN2*-Forward: GAGAGTGAGCGGCAGAGC; *AXIN2*-Reverse: CGGCTGACTCGTTCTCCT; *GAPDH*-Forward: TCAAGAAGGTGGTGAAGCAGG; *GAPDH*-Reverse: ACCAGGAAATGAGCTTGACAAA.

## Western blotting

Western blotting was performed as described previously (Ertych et al, 2014) using the following primary antibodies: anti-ATR (rabbit, 1:1,000, Cat# 2790, RRID:AB_2227860; Cell Signaling Technology), anti-phospho-ATR (rabbit, 1:1,000, Cat# GTX128145, RRID: AB_2687562; GeneTex), anti-Chk1 (mouse, 1:1,000, Cat# 2360, RRID: AB_2080320; Cell Signaling Technology), anti-phospho-Chk1 (rabbit, 1:2,000, Cat# 2349, RRID:AB_2080323; Cell Signaling Technology), and anti-GAPDH (mouse, 1:3,000, Cat# sc-365062, RRID: AB_10847862; Santa Cruz Biotechnology). The secreted Wnt10b protein was detected by Western blotting upon enrichment using Blue Sepharose beads (17-0948-01; Th. Geyer Inc.). Enriched Wnt10b was resolved on 10% SDS–PAGE gels and detected by Western blotting using anti-Wnt10b antibodies (MABN717, 1:1,000, 5A7, RRID: AB_3675944; Merck-Millipore). Anti-HSC70 antibodies (#sc-7298, 1: 1,000, B-6, RRID: AB_627761; Santa Cruz) were used as a loading control.

## Cell cycle analysis

Cell cycle distribution and mitotic content were determined by FACS analysis using a BD FACSCanto II (BD Biosciences) and analyzed using FACSDiva software (version 6; BD Biosciences), as described previously (Ertych et al, 2014).

## EdU FACS

EdU incorporation was analyzed as a measure of replication progression as described previously (Macheret & Halazonetis, 2019). In brief, cells were pulse-labeled with 20 $\mu$M EdU for 30 min, washed 5 times with PBS, and fixed with 70% ethanol. The cells were washed in PBS, and EdU was labeled with Alexa Fluor 488 azide (Invitrogen) by an EdU Click-iT reaction (reagents for 1 ml: 855 $\mu$l of 0.1 mM Tris–HCl, pH 8.0, 40 $\mu$l of CuSO4, 100 $\mu$l of 1 M sodium L-ascorbate) and incubated in 500 $\mu$l of 1 mg/ml DNase-free RNase A in PBS for 30 min at RT. DNA was stained with 1 $\mu$g/ml propidium iodide. EdU incorporation was measured using a BD FACSCanto II (BD Biosciences) and analyzed using FACSDiva software (version 6; BD Biosciences).

## Measurements of microtubule plus–end growth rates

To measure microtubule plus–end assembly rates, comets of fluorescently labeled microtubule end–binding protein 3 (EB3) were tracked by live-cell fluorescence microscopy at 37°C/5% $CO_2$. Cells were transfected with pEGFP-EB3 (kindly provided by Linda Wordeman) or pcDNA3-EB3-StayGold (kindly provided by Atsushi Miyawaki) (Hirano et al, 2022) plasmids and analyzed using a Delta Vision Elite live-cell microscope equipped with a PCO Edge sCMOS camera and a PlanApo N 60x/1.42 Oil ∞/0.17/FN26.5 objective as previously described (Ertych et al, 2014; Bohly et al, 2022). For mitotic measurements, cells were accumulated in mitosis by treatment with 2 $\mu$M dimethylenastron (#SML0905; Sigma-Aldrich). Average assembly rates were calculated from 25 individual measurements per cell, and a total of 30 cells from three independent experiments were analyzed per condition.

## Cell synchronization

To synchronize cells at the G1/S transition of the cell cycle, cells were synchronized by a double thymidine block protocol as described previously (Schmidt et al, 2021). 2 mM thymidine (#sc-296542A; Santa Cruz) was added for 16 h. Cells were released into fresh culturing medium by washing with fresh culturing medium every 5 min for 30 min, cultured for 8 h, and subjected to a second thymidine block for 16 h before releasing from the G1/S block. Cells were further analyzed at different time points after the release.

## Treatment of cell cycle–synchronized cells

Cell cycle phase–specific treatments (Fig 2) were done using cells synchronized at the G1/S transition using a double thymidine block protocol. Cells were released for various times as indicated in Fig 2A and treated with DKK1, GSK3 inhibitor, Wnt3a, or Wnt10b for 2 h followed by washout of the reagents. Treatments specific to the early S phase (Fig 3) were achieved by release from G1/S synchronization for 2 h followed by washout of the treatments. Induction of replication stress in the early S phase (Figs 5 and 6) was achieved in the same manner, whereas treatment with hydroxyurea was done 2 h after the release from the G1/S block to avoid cell cycle arrest before the cell enters the S phase. Hydroxyurea was washed out 4 h later.

## Detection of chromosome missegregation

The appearance of lagging chromosomes during anaphase was used as a measure for chromosome missegregation as described

previously (Cimini et al, 2001). Cells grown on glass coverslips were enriched by cell cycle synchronization using a double thymidine block and release protocol, and fixed with 2% p-formaldehyde/PBS followed by treatment with ice-cold methanol at –20°C for 5 min. Mitotic spindles, kinetochores, and chromosomes were detected by immunofluorescence microscopy using anti-$\alpha$-tubulin antibodies (1:700, sc-23948, RRID:AB_628410; Santa Cruz), anti-Cenp-C antibodies (1:1,000, PD030, RRID:AB_10693556; MBL), and Hoechst 33342 (#H3570; RRID:AB_3675235; Thermo Fisher Scientific). Images were captured at RT using a Leica DMI6000B microscope (Leica) equipped with a Leica DFC360 FX camera, HCX PL APO 63x/1.30 objective, and Leica LAS-AF software (Leica). A lagging chromosome was defined as Cenp-C–positive DNA clearly separated from the polar DNA masses in late anaphase cells.

### Detection of chromosomal breaks

Chromosomal breaks were detected on metaphase chromosome spreads. Cells were treated with 2 $\mu$M dimethylenastron for 3 h to enrich mitotic cells in prometaphase. Cells were harvested, resuspended in 40% (vol/vol) RPM1-1640 medium (PAN Biotech)/60% $H_2O$, and fixed in 75% methanol/25% glacial acetic acid. Cells were suspended in 100% glacial acetic acid and dropped onto pre-cooled microscopy slides. The slides were dried, and DNA was stained with Giemsa solution (AppliChem). Bright-field microscopy was performed at RT using a Leica DM IL LED microscope (Leica) equipped with a Leica HI Plan 63x/0.75 ∞/0.17 objective and an ODC832 camera (Kern & Sohn GmbH) to detect and to quantify the appearance of chromosomes with visible breaks.

### Molecular DNA combing

Molecular DNA combing was performed to determine DNA replication fork progression rates and inter-origin distances as a measure for activation of additional origins (Moore et al, 2022). Newly synthesized DNA was sequentially labeled with 100 $\mu$M 5-chloro-2′-deoxyuridine (CldU; #C6891; Sigma-Aldrich) and 100 $\mu$M 5-iodo-2′-deoxyuridine (IdU; #I7125; Sigma-Aldrich) for 30 min each. Cells were harvested and processed using the FiberPrep DNA extraction kit (Genomic Vision) according to the manufacturer's protocol. The isolated and purified ssDNA was combed on salinized microscopy slides (Genomic Vision) using Molecular Combing System (Genomic Vision). Combed DNA samples were stained with primary anti-CldU antibodies (#ab6326, 1:10, BU1/75, ICR1, RRID: AB_305426; Abcam), anti-IdU antibodies (#347580, 1:10, B44, RRID: AB_10015219; BD Biosciences), anti-ssDNA antibodies (#autoanti-ssDNA, 1:5, RRID:AB_10805144; DSHB), and secondary antibodies conjugated to Alexa Fluor 488 (A21121, 1:25, RRID:AB_2535764; Invitrogen), Alexa Fluor 594 (#150160, 1:25, RRID:AB_2756445; Abcam), and BV421 (#563846, 1:25, RRID:AB_2738449; BD Biosciences). Labeled DNA fibers were imaged at RT using a Delta Vision Elite microscope (Delta Vision) equipped with a PlanApo N 60x/1.42 Oil ∞/0.17/FN26.5 objective and a PCO edge sCMOS camera (PCO). Labeled stretches of DNA were analyzed to calculate fork speeds and inter-origin distances as described previously (Bohly et al, 2022).

### Statistical analysis

Statistical analysis was performed, and graphs were drawn using GraphPad Prism 9.0 software (GraphPad Software). Mean values and SD were calculated for each experiment. Microtubule polymerization rate measurements and DNA combing analyses were depicted as box plots, where the whiskers represent the 10th and 90th percentile, the boxes the 25th and 75th percentile, the line in the boxes the median value, and the plus sign within the boxes the average value. All experimental results are based on at least three independently performed biological replicates as indicated in the figure legends. Statistical analysis was performed using unpaired two-tailed $t$ tests. The statistical analysis of the Wnt/$\beta$-catenin signaling activity was performed using an unpaired one-sample $t$ test using Microsoft Excel. Significances for experiments are indicated as $P$-values: ns (not significant), $P \geq 0.05$, *$P < 0.05$, *$P < 0.01$, ***$P < 0.001$.

# Data Availability

All raw data generated in this study will be made available upon request.

# Supplementary Information

# Acknowledgements

We thank Atsushi Miyawaki, Feng Zhang, Zuzana Storchova, and Linda Wordeman for providing plasmids and cell lines. This work was funded by the Deutsche Forschungsgemeinschaft (DFG), project number 331351713 – SFB1324 (to H Bastians, G Davidson, M Boutros, and SP Acebron).

## Author Contributions

A Haas: data curation, formal analysis, validation, investigation, visualization, methodology, and writing—original draft, review, and editing.
F Wenz: data curation, formal analysis, validation, investigation, visualization, methodology, and writing—review and editing.
J Hattemer: formal analysis, investigation, methodology, and writing—review and editing.
J Wesslowski: formal analysis, investigation, methodology, and writing—review and editing.
G Davidson: conceptualization, resources, data curation, formal analysis, funding acquisition, and writing—review and editing.
O Voloshanenko: data curation, formal analysis, investigation, methodology, and writing—review and editing.
M Boutros: resources, supervision, funding acquisition, and writing—review and editing.
SP Acebron: conceptualization, resources, data curation, funding acquisition, methodology, and writing—review and editing.

H Bastians: conceptualization, resources, data curation, formal analysis, supervision, funding acquisition, validation, visualization, project administration, and writing—original draft, review, and editing.

## Conflict of Interest Statement

The authors declare that they have no conflict of interest.

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
