## [Reviewer comments · Life Science Alliance]

Life Science Alliance

Wnt10b signaling regulates replication stress-induced chromosomal instability in human cancer

Alexander Haas, Friederike Wenz, Janina Hattemer, Janine Wesslowski, Gary Davidson, Oksana Voloshanenko, Michael Boutros, Sergio Acebron, and Holger Bastians

DOI: <https://doi.org/10.26508/lsa.202503295>

Corresponding author(s): *Holger Bastians, Universitätsmedizin Göttingen*

Review Timeline:

Submission Date:	2025-03-06
Editorial Decision:	2025-04-23
Revision Received:	2025-06-24
Editorial Decision:	2025-07-22
Revision Received:	2025-08-04
Accepted:	2025-08-08

Scientific Editor: Sarita Hebbar

Transaction Report:

April 23, 2025

Re: Life Science Alliance manuscript #LSA-2025-03295

Prof. Holger Bastians
Universitätsmedizin Göttingen
Department of Molecular Oncology
Grisebachstrasse 8
Göttingen 37077
Germany

Dear Dr. Bastians,

Thank you for submitting your manuscript entitled "Wnt10b signaling regulates replication stress-induced chromosomal instability in human cancer" to Life Science Alliance (LSA). The manuscript was assessed by three expert reviewers, whose comments are appended to this letter.

Overall, the three reviewers agreed that this work is of potential significance to the field. That said, we agree with the reviewers that some important aspects of the manuscript need to be addressed for publication at LSA. These are:

1. New experiments and/or existing data to be reassessed and presented:
 - Assess if MT polymerisation rates are altered upon wash off of the GSK3i or Taxol (Reviewer 2, point 1)
 - Assess the effect of APH treatment (replication stress) on the Wnt signaling pathway (Reviewer 3, paragraph 4)
 - Assess the differences following DKK1 (Wnt inhibition) treatment on CIN- and CIN+ colorectal cancer cells (Reviewer 2, point 3 & Reviewer 3, paragraph 4)
 - Include snapshots of the images of microtubules and/or lagging chromosome at key stages through mitosis (Reviewer 2, paragraph 1)
2. Aspects of the results must be further elaborated upon, with changes to the text, namely
 - other sources of stress and other effects of nucleoside supplementation (Reviewer 1, paragraph 3),
 - the underlying mechanism for the distinct effects of Wnt10b and Wnt3a (Reviewer 2, point 4) on MT dynamics
 - on differences in cell cycle observations (Reviewer 3, paragraph 7)

In view of these recommendations, we invite you to submit a revised manuscript addressing the reviewers' comments. When submitting the revision, please include a letter addressing all the reviewers' comments point by point. While a rebuttal must respond to all points in some form, additional data to resolve these points may not be required.

Thank you for this interesting contribution to Life Science Alliance. We are looking forward to receiving your revised manuscript.

Sincerely,

Sarita Hebbar, PhD
Scientific Editor
Life Science Alliance
<http://www.lsjournal.org>

B. MANUSCRIPT ORGANIZATION AND FORMATTING:

Reviewer #1 (Comments to the Authors (Required)):

The manuscript "Wnt10b signaling regulates replication stress-induced chromosomal instability in human cancer" by Haas et. al is focused on the role of Wnt10b signaling in protecting cells from genome instability caused by replication stress. The authors show that replication stress leads to an increase in the rate of microtubule growth resulting in genome instability, that can be prevented by Wnt10b signaling or exacerbated by its loss. Interestingly, the microtubule dynamics are regulated in S-phase, which affects the accuracy of mitotic segregation later in the cell cycle. The manuscript is generally well written, the findings are novel and well-supported by the presented data. I believe the paper can be published after a few minor adjustments.

Replication stress is tested by DNA combing, which only indicates fork speed/inter-origin distance. The hallmark of replication stress is the accumulation of ssDNA and activation of the checkpoint kinases ATR/CHK1. Using one of these markers to indicate the presence of absence of replication stress would strengthen the conclusions.

Using nucleoside supplementation to mitigate replication stress (Fig 4C-D) is an imperfect approach. Nucleosides may have other effect on the cells, unrelated to DNA replication - including effects of metabolism and signaling (G-proteins). The authors should discuss the other possibilities and consider alternative approaches. However, I believe the overall conclusions of the paper are well-supported even without the nucleoside experiments.

Reviewer #2 (Comments to the Authors (Required)):

The central findings from this study are truly intriguing and provocative suggesting that Wnt signalling induces CIN via replication stress in cancers. It would be useful to include snapshots of the images of microtubules and/or lagging chromosomes during some of the key events in the cells as they progress through mitosis in each of the figures. This would provide a much better appreciation for the data as opposed to only the graph in each of the figures.

1. Figure 1C: Can the increased growth rates of the microtubules purely be attributed to the inhibition of Wnt signalling, or do these inhibitors also show any off-target effects on cell shape and cell proliferation, which in turn impacts MT polymerization rates? Also, were the MT rates altered upon wash off of the GSK3i or Taxol? The data in Fig 1D nicely shows the extent of lagging chromosomes. This data is further interesting considering the specificity of the effects of Wnt10b as Wnt3b. Are there any specific shared sub-domains between these proteins that would contribute to altered MT dynamics? I am assuming that interacting partners of Wnt10b would also show an overarching effect on MT dynamics and regulation?

2. Figure:2 This data is particularly interesting in that the inhibition of Wnt signalling within the 2 hour time window of the S-

phase determine mitotic error rates in cancer cells. Is there any effect of prolonging S-phase on MT dynamics and error rates? Also, is the extent of lagging chromosomes particularly noisy in late S and G2 phase? what contributes to this noise?

3. Figure 3: Is it reasonable to conclude that the effects of DKK1 or EVI show a cytoskeletal memory in cells, as proposed? does this differ between CIN- and CIN+ colorectal cancer cells? in other words, would cytoskeletal memory be compromised in CIN+ and aggressive cancers as opposed to CIN- and less aggressive cancers?

4. While the study overall brings to the forefront certain intriguing aspects of Wnt signalling and its role on regulating MT dynamics in CIN- and CIN+ cells, the study could delve more on the underlying mechanistic aspects of how Wnt signalling and especially the effects of Wnt10b and not Wnt3a?

Reviewer #3 (Comments to the Authors (Required)):

Replication stress (RS) is a well-established driver of chromosomal instability (CIN), promoting DNA under-replication and mitotic abnormalities. Under-replicated DNA leads to chromosome breakage and rearrangements, as well as chromosome non-disjunction, resulting in both structural and numerical chromosomal abnormalities. Additionally, a growing body of evidence, including previous work by the authors, has shown that DNA replication stress induces whole-chromosome mis-segregation through alterations in microtubule (MT) dynamics and spindle assembly. The authors have previously characterized the role of the Wnt signaling pathway in regulating MT dynamics, showing that inhibition of Wnt signaling increases MT growth rates and chromosome missegregation. They also demonstrated that Wnt signaling can influence DNA replication in pluripotent stem cells. Here, using chromosomally stable (HCT116) or unstable (CIN+) colorectal cancer cell lines, they show that Wnt10b signaling acts during S-phase to prevent both MT alterations and RS-associated chromosome breakage.

While the findings linking RS with mitotic defects through Wnt signaling are of interest, they appear mostly correlative, and the underlying mechanism has not been investigated. Additionally, some inconsistencies should be addressed to support the authors' conclusions:

- The authors show that both RS and Wnt inhibition affect MT dynamics, suggesting that RS mimics Wnt inhibition. They assessed whether Wnt inhibition impacts RS and state that it does not; however, results in Figure 4F and Supplementary Figure S3 show that Wnt10b significantly increases replication fork speed in SW480 CIN+ cancer cells and in APH-treated RPE1-hTert non-cancerous cells, with a concomitant increase in inter-origin distance. Therefore, the data support the conclusion that Wnt signaling modulates RS, which is consistent with what the authors previously reported in pluripotent stem cells. In addition, it is unclear why the RPE1 cells were used for experiments in Fig. S3.

- Since RS mimics Wnt inhibition, did the authors assess the effect of RS on the Wnt signaling pathway? What is the effect of APH treatment on Wnt signaling? Is Wnt signaling inhibited in CIN+ cells? These questions should be addressed to clarify the relationship between RS, Wnt signaling, and MT dynamics.

- Related to the previous point, the results in Figure 6 show that Wnt10b reduces chromosome breaks in HCT116 cells treated with RS-inducing agents or knocked out for the Wnt secretion factor EVI, as well as in CIN+ colorectal cancer cells. Based on previous results, this reduction could be due to the alleviation of RS. Likewise, is the endogenous RS observed in CIN+ cells caused by defective Wnt secretion/signaling? This possibility should be investigated to further support the authors' conclusions.

Minor Points:

- Supplementary Figure S1: The Wnt reporter assay should be performed in the same cell lines used in the other experiments.

- Supplementary Figure S2: Although the cell cycle profile appears similar between parental and EVI-/- HCT116 cells, the DNA content seems shifted, particularly at 6 hours (G2 phase). Could the authors explain why this is? Additionally, the FACS analysis of cell cycle progression in Supplementary Figure S3 shows reduced M-phase entry in EVI-/- HCT116 cells, which may be due to RS associated with Wnt inhibition. A more detailed analysis of the cell cycle, including BrdU or EdU incorporation, would help better evaluate the impact of Wnt signaling during the S and G2-M phases.

Rebuttal letter for the revised manuscript entitled “Wnt10b signaling regulates replication stress-induced chromosomal instability in human cancer”, by Alexander Haas et. al. (Life Science Alliance manuscript #LSA-2025-03295)

List of newly added data/figures:

1. Figure 1D: additional example images of cells showing chromosome segregation
2. Figure 1: movie of microtubule plus end tracking (supplemental movie 1)
3. Figure 2G: example images of cells showing chromosome segregation
4. Figure 3B: example images of cells showing interphase microtubule plus ends
5. Figure 4B: example images of cells showing chromosome segregation
6. Figure 4D: example images of cells showing chromosome segregation
7. Figure 5: example images of cells showing chromosome segregation
8. Figure S1B: additional Wnt reporter assays using HCT116 and RPE1-hTert cells
9. Figure S2B: microtubule growth rates with and without GSK3 inhibitor washout
10. Figure S3A: microtubule growth rates in CIN+ cells after additional DKK1 treatment
11. Figure S3B: chromosome missegregation in CIN+ cells after additional DKK1 treatment
12. Figure S4A: EdU-FACS analysis of cells treated with aphidicolin or DKK1
13. Figure S4C: western blots detecting activated ATR and CHK1 kinases
14. Figure S5A: FACS analyses of asynchronously growing cells +/- GSK3b inhibitor or DKK1
15. Figure S5B: FACS analyses of synchronized cells +/- DKK1 and analysis of mitotic entry
16. Figure S6A: Wnt activity (qPCR) upon induction of replication stress in HCT116 cells
17. Figure S6B: Wnt activity (qPCR) upon alleviation of replication stress in CIN+ cancer cells

Text changes made in the Manuscript are marked in red.

Rebuttal to the Comments from the editor

Overall, the three reviewers agreed that this work is of potential significance to the field. That said, we agree with the reviewers that some important aspects of the manuscript need to be addressed for publication at LSA. These are:

1. New experiments and/or existing data to be reassessed and presented:

- Assess if MT polymerisation rates are altered upon wash off of the GSK3i or Taxol (Reviewer 2, point 1)

We performed additional experiments using EVI/WIs knockout cells treated with GSK3i during S phase followed by washout of the drug and measurements of microtubule growth rates in mitosis. As expected, GSK3i treatment in S phase rescued increased MT dynamics and wash-off of the drug did not change this result. This result is expected since we demonstrate an S phase specific role of Wnt-GSK3 signaling. In further support of this, GSK3i treatment of cells after S phase does not rescue increased MT polymerization rates in mitosis. Thus, GSK3 inhibition has to be present in S phase to allow rescue of the phenotype and subsequent wash-off has no effect. These additional experiments are now presented in Fig. S2B.

- Assess the effect of APH treatment (replication stress) on the Wnt signaling pathway (Reviewer 3, paragraph 4)

We analyzed canonical Wnt signaling activity by assessing beta-catenin-mediated transcriptional induction of its target gene AXIN2 in response to replication stress induction and upon alleviation of replication stress in CIN+ cancer cells. We found no alterations in Wnt activity in both settings. We show these results in Fig. S6.

- Assess the differences following DKK1 (Wnt inhibition) treatment on CIN- and CIN+ colorectal cancer cells (Reviewer 2, point 3 & Reviewer 3, paragraph 4)

We performed additional experiments and analyzed the effects of DKK1 treatment on HCT116 (CIN-) and SW480 and SW620 (CIN+) cells. The increased microtubule growth rates and high chromosome missegregation rates generally found in CIN+ were not further increased upon Wnt inhibition supporting a role of Wnt10b signaling downstream of replication stress. These results are now presented in Fig. S3.

- Include snapshots of the images of microtubules and/or lagging chromosome at key stages through mitosis (Reviewer 2, paragraph 1)

As suggested, we include additional example images of cells exhibiting chromosome missegregation in Figs. 1D, 2G, 4B, 4D and 5C. In addition, we now provide an example movie on the measurement of microtubule growth rates in mitosis (see Supplementary movie S1) and we provide example images of cells showing GFP-EB3-labelled interphase microtubule plus ends (Figure 3B).

2. Aspects of the results must be further elaborated upon, with changes to the text, namely
- other sources of stress and other effects of nucleoside supplementation (Reviewer 1, paragraph 3),

We now included a statement on nucleoside supplementation (page 8: "Although deoxynucleoside supplementation might affect microtubule dynamics and mitosis also by other yet not investigated mechanisms, e.g. by affecting purinergic signaling, these results strongly suggest that DNA replication stress present in chromosomally unstable cancer cells causes abnormal mitotic microtubule dynamics and chromosome missegregation as demonstrated previously"). Please see also the Rebuttal to Reviewer #1.

Also, we discuss the possibility that other types of stress signaling, e.g. upon DNA damage, might also be regulated by Wnt10b signaling (see Discussion section, page 13)

- the underlying mechanism for the distinct effects of Wnt10b and Wnt3a (Reviewer 2, point 4) on MT dynamics

We now include a discussion on the distinct effects of Wnt10b vs. Wnt3a on the regulation of MT dynamics (see discussion section, page 14).

- on differences in cell cycle observations (Reviewer 3, paragraph 7)

To clarify this point, we performed additional cell cycle analyses, which are now shown in Fig. S5. Please also refer to the information given below to Reviewer #3.

Rebuttal to Reviewer #1:

The manuscript "Wnt10b signaling regulates replication stress-induced chromosomal instability in human cancer" by Haas et. al is focused on the role of Wnt10b signaling in protecting cells from genome instability caused by replication stress. The authors show that replication stress leads to an increase in the rate of microtubule growth resulting in genome instability, that can be prevented by Wnt10b signaling or exacerbated by its loss. Interestingly, the microtubule dynamics are regulated in S-phase, which affects the accuracy of mitotic segregation later in the cell cycle. The manuscript is generally well written, the findings are novel and well-supported by the presented data. I believe the paper can be published after a few minor adjustments.

We thank the reviewer for the overall very positive evaluation of our work.

Replication stress is tested by DNA combing, which only indicates fork speed/inter-origin distance. The hallmark of replication stress is the accumulation of ssDNA and activation of the checkpoint kinases ATR/CHK1. Using one of these markers to indicate the presence of absence of replication stress would strengthen the conclusions.

It is correct that experimentally induced replication stress (e.g. using aphidicolin) activates ATR-Chk1 checkpoint signaling, which is responsible to halt the cell cycle during S or G2/M in order to allow fork repair and fork restart. However, this is only the case upon moderate to severe replication stress. As we showed previously (Böhly, ..., Bastians, 2019), mild replication stress that is typically present in chromosomally unstable cancer cells and which can be mimicked by low concentrations of aphidicolin (100 nM) escape this checkpoint control and does not activate ATR-Chk1 signaling and thus, does not trigger cell cycle arrest. This allows entry into mitosis in the presence of (unrecognized) mild replication stress and causes mitotic errors. In our work presented here, using DNA combing analysis as a highly sensitive measure to detect also mild replication stress, we did not detect replication stress upon Wnt inhibition.

Nevertheless, as suggested by the reviewer, we performed additional experiments detecting phosphorylated (activated) ATR and Chk1 kinases on western blots, (a) upon mild and moderate replication stress induced by 100 and 200 nM aphidicolin, respectively, and (b) upon DKK1-mediated Wnt inhibition. As expected, only aphidicolin-mediated replication stress, but not Wnt inhibition caused increased phosphorylation and thus, activation of ATR and Chk1. These results are now presented in Fig. S4C.

In this regard, we also performed additional analyses on cell cycle progression upon Wnt inhibition. We did not observe any significant changes in cell cycle profiles or entry into mitosis upon Wnt inhibition showing that Wnt inhibition is not associated with S phase delay (see Figure S5).

Using nucleoside supplementation to mitigate replication stress (Fig 4C-D) is an imperfect approach. Nucleosides may have other effect on the cells, unrelated to DNA replication - including effects of metabolism and signaling (G-proteins). The authors should discuss the other possibilities and consider alternative approaches. However, I believe the overall conclusions of the paper are well-supported even without the nucleoside experiments.

We agree with the reviewer that addition of nucleosides may also affect other signaling pathways. It is of note that we used deoxynucleosides and not nucleosides, which represents a well-accepted mean to alleviate replication stress. Nevertheless, to make this point clearer, we included a statement on page 8: "Although deoxynucleoside supplementation might affect microtubule dynamics and mitosis also by other yet not investigated mechanisms, e.g. by affecting purinergic signaling, these results strongly suggest that DNA replication stress present in chromosomally unstable cancer cells causes abnormal mitotic microtubule dynamics and chromosome missegregation as demonstrated previously."

Rebuttal to Reviewer #2:

The central findings from this study are truly intriguing and provocative suggesting that Wnt signalling induces CIN via replication stress in cancers. It would be useful to include snapshots of the images of microtubules and/or lagging chromosomes during some of the key events in the cells as they progress through mitosis in each of the figures. This would provide a much better appreciation for the data as opposed to only the graph in each of the figures.

We are grateful to reviewer #2 for acknowledging the importance of our work.

In addition to the quantification of chromosome missegregation we are now providing additional example images in Figures 1D, 2G, 4B, 4D and 5C.

In addition, we provide an example movie of a cell used for live cell microtubule plus end tracking live cell measurements (Supplemental movie 1). Also, we provide snap shots of cells used for interphase microtubule plus end tracking (Figure 3B).

1. Figure 1C: Can the increased growth rates of the microtubules purely be attributed to the inhibition of Wnt signalling, or do these inhibitors also show any off-target effects on cell shape and cell proliferation, which in turn impacts MT polymerization rates?

In our study, we used several means to inhibit Wnt(10b) signaling: (a) DKK treatment, (b) knockout of EVI/Wls, (c) knockout of Wnt10B. Vice versa, we use recombinant purified Wnt3 and Wnt10b proteins and established GSK3 inhibitor to activate Wnt signaling. All treatments result in consistent and robust effects on microtubule dynamics. Therefore, we think that the effects on microtubules are due to Wnt inhibition. Based on our cell cycle analyses we provided (Fig. S5) we have no indication for a change in proliferation/cell cycle progression upon Wnt modulation.

Also, were the MT rates altered upon wash off of the GSK3i or Taxol?

As suggested, we performed additional experiments in which we treated cells in S phase with GSKi followed by wash-off of the inhibitor and MT growth rates measurements in mitosis. As expected, GSK inhibition during S phase is sufficient to rescue the increased MT growth rates in mitosis and wash-off of the drug has no effect. In addition, treatment with GSKi after S phase (in G2) does not rescue increased MT growth rates re-emphasizing the role of Wnt-GSK signaling solely in S phase. These data are presented in Fig. S2B.

The data in Fig 1D nicely shows the extent of lagging chromosomes. This data is further interesting considering the specificity of the effects of Wnt10b as Wnt3b. Are there any specific shared sub-domains between these proteins that would contribute to altered MT dynamics? I am assuming that interacting partners of Wnt10b would also show an overarching effect on MT dynamics and regulation?

We are now providing a discussion part on our intriguing finding that Wnt10b, but not Wnt3a is acting downstream of replication stress to regulate mitotic chromosome segregation (see discussion section, page 14)

2. Figure 2: This data is particularly interesting in that the inhibition of Wnt signaling within the 2 hour time window of the S-phase determine mitotic error rates in cancer cells. Is there any effect of prolonging S-phase on MT dynamics and error rates? Also, is the extent of lagging chromosomes particularly noisy in late S and G2 phase? what contributes to this noise?

Since neither mild replication stress nor Wnt inhibition result in a delay in S phase (see Supplementary Fig S5) we think that a delay in S phase per se is not key for triggering abnormally

increased microtubule grow rates.

3. Figure 3: Is it reasonable to conclude that the effects of DKK1 or EVI show a cytoskeletal memory in cells, as proposed? does this differ between CIN- and CIN+ colorectal cancer cells? in other words, would cytoskeletal memory be compromised in CIN+ and aggressive cancers as opposed to CIN- and less aggressive cancers?

Our results demonstrate that CIN- cancer cells exhibit proper MT growth rates from S phase until mitosis. Increased MT dynamics can be triggered in these cells by inducing replication stress or by Wnt inhibition (e.g. DKK treatment) only during S phase. In contrast, CIN+ show increased MT growth rates from S phase until mitosis and this is due to endogenous replication stress in these cells. We think that the term “cytoskeletal memory” might be misleading and we omitted it now from the text.

As suggested also by the editor, we tested whether Wnt inhibition has additional effects on MT dynamic changes in CIN- and CIN+ cells. While DKK1 treatment increased microtubule dynamics and chromosome missegregation in non-CIN cells as shown before, CIN+ cells suffering from endogenous replication stress show no further increase in mitotic errors supporting a role of Wnt signaling downstream of replication stress. These data are now shown in Fig. S3A and B.

4. While the study overall brings to the forefront certain intriguing aspects of Wnt signalling and its role on regulating MT dynamics in CIN- and CIN+ cells, the study could delve more on the underlying mechanistic aspects of how Wnt signalling and especially the effects of Wnt10b and not Wnt3a?

We now include more discussion on possible mechanistic aspects on the role of Wnt10b vs. Wnt3a signaling in the regulation of CIN (see discussion section, page 14).

Rebuttal to Reviewer #3:

Replication stress (RS) is a well-established driver of chromosomal instability (CIN), promoting DNA under-replication and mitotic abnormalities. Under-replicated DNA leads to chromosome breakage and rearrangements, as well as chromosome non-disjunction, resulting in both structural and numerical chromosomal abnormalities. Additionally, a growing body of evidence, including previous work by the authors, has shown that DNA replication stress induces whole-chromosome mis-segregation through alterations in microtubule (MT) dynamics and spindle assembly. The authors have previously characterized the role of the Wnt signaling pathway in regulating MT dynamics, showing that inhibition of Wnt signaling increases MT growth rates and chromosome missegregation. They also demonstrated that Wnt signaling can influence DNA replication in pluripotent stem cells. Here, using chromosomally stable (HCT116) or unstable (CIN+) colorectal cancer cell lines, they show that Wnt10b signaling acts during S-phase to prevent both MT alterations and RS-associated chromosome breakage.

While the findings linking RS with mitotic defects through Wnt signaling are of interest, they appear mostly correlative, and the underlying mechanism has not been investigated. Additionally, some inconsistencies should be addressed to support the authors' conclusions:

- The authors show that both RS and Wnt inhibition affect MT dynamics, suggesting that RS mimics Wnt inhibition. They assessed whether Wnt inhibition impacts RS and state that it does not; however, results in Figure 4F and Supplementary Figure S3 show that Wnt10b significantly increases replication fork speed in SW480 CIN+ cancer cells and in APH-treated RPE1-hTert non-cancerous cells, with a concomitant increase in inter-origin distance. Therefore, the data support the conclusion that Wnt signaling modulates RS, which is consistent with what the authors previously reported in pluripotent stem cells. In addition, it is unclear why the RPE1 cells were used for experiments in Fig. S3.

The differences in replication fork speed for EVI KO cells and for SW480 cells shown in Fig. 4F are statistically (little) significant, but very subtle. However, we observe no significant differences for cells treated with DKK1 or in SW620 cells (see Fig. 4F) indicating that Wnt inhibition does not consistently regulate replication stress in somatic cells. We used RPE1 cells as an example for non-cancer cells (Fig. S4B).

- Since RS mimics Wnt inhibition, did the authors assess the effect of RS on the Wnt signaling pathway? What is the effect of APH treatment on Wnt signaling? Is Wnt signaling inhibited in CIN+ cells? These questions should be addressed to clarify the relationship between RS, Wnt signaling, and MT dynamics.

As suggested by the reviewer, we determined (canonical) Wnt activity by quantifying the expression of the bona fide Wnt target gene AXIN2 by qPCR analysis. We found no change in AXIN2 expression neither upon the induction of replication stress (aphidicolin treatment) nor upon alleviation of (endogenous) replication stress in CIN+ cancer cells. Thus, replication stress per se seems not to impact on canonical, beta-catenin-dependent Wnt activity. These results are now presented in Figure S6).

However, we are currently not able to measure Wnt10b activity due to the fact that the relevant target of Wnt10b signaling is yet unidentified. In fact, we show that Wnt10b, in contrast to Wnt3a, does hardly activate canonical Wnt signaling (Figure S1B). We propose that Wnt10b acts as part of Wnt/STOP signaling, possibly leading to phosphorylation and/or stabilization of one or more proteins involved in the regulation of microtubule dynamics (see Discussion section, page 14). We

also discuss now the possibility that CIN+ cancer cells may suffer from impaired Wnt10b signaling (see discussion section, page 13).

- Related to the previous point, the results in Figure 6 show that Wnt10b reduces chromosome breaks in HCT116 cells treated with RS-inducing agents or knocked out for the Wnt secretion factor EVI, as well as in CIN+ colorectal cancer cells. Based on previous results, this reduction could be due to the alleviation of RS. Likewise, is the endogenous RS observed in CIN+ cells caused by defective Wnt secretion/signaling? This possibility should be investigated to further support the authors' conclusions.

Our study showed that Wnt10b signaling rescues mitotic errors and chromosomal breaks upon replication stress. However, our DNA combing analyses (Fig. 4F) showed that endogenous replication stress present in CIN+ cancer cells is not (or hardly, see above) alleviated by Wnt10b activation, strongly suggesting that the endogenous replication stress in these cancer cells is not due to loss of Wnt10b signaling. We discuss the possibility that Wnt10b signaling might be altered in CIN+ cancer cells (see Discussion section, page 13).

Minor Points:

- Supplementary Figure S1: The Wnt reporter assay should be performed in the same cell lines used in the other experiments.

We performed additional Wnt reporter assays using HCT116 and RPE1-hTert cells, together validating that Wnt10b, in contrast to Wnt3a, hardly activates canonical Wnt signaling (Fig. S1B).

- Supplementary Figure S2: Although the cell cycle profile appears similar between parental and EVI-/- HCT116 cells, the DNA content seems shifted, particularly at 6 hours (G2 phase). Could the authors explain why this is? Additionally, the FACS analysis of cell cycle progression in Supplementary Figure S3 shows reduced M-phase entry in EVI-/- HCT116 cells, which may be due to RS associated with Wnt inhibition. A more detailed analysis of the cell cycle, including BrdU or EdU incorporation, would help better evaluate the impact of Wnt signaling during the S and G2-M phases.

The representative FACS profiles (Fig. S2A) are only examples from several experiments we performed. FACS analysis often show some minor variations. We think that the very slight shift after 6 hours after thymidine release is not very relevant. No differences are seen at t=4 or at t=8 hour.

In the FACS analysis now shown in Fig. S5C, EVI knockout cells showed a slightly accelerated entry into mitosis (not delayed).

To further clarify this, we performed additional FACS-bases analysis on asynchronously growing cells (Fig. S5A) and on synchronized cells treated with DKK1 (Fig. S5B). In all cases we did not observe any significant differences in cell cycle timing or in mitotic entry rates upon Wnt inhibition. In addition, as suggested by this reviewer, we also performed EdU incorporation assays, which showed reduced EdU incorporation in S phase upon aphidicolin treatment as expected, but not after DKK1-mediated Wnt inhibition (Fig. S4A).

July 22, 2025

RE: Life Science Alliance Manuscript #LSA-2025-03295R

Prof. Holger Bastians
Universitätsmedizin Göttingen
Department of Molecular Oncology
Grisebachstrasse 8
Göttingen 37077
Germany

Dear Dr. Bastians,

Thank you for submitting your revised manuscript entitled "Wnt10b signaling regulates replication stress-induced chromosomal instability in human cancer". It was evaluated by two of the original reviewers whose comments are appended below.

Based on their assessment, we would be happy to publish your paper in Life Science Alliance pending final revisions necessary to meet our formatting guidelines.

- We encourage the author to head to a minor suggestion for Figure S4B (previously Figure S3A) from Reviewer 3.
- Please provide information on temperature of stage/slide during image acquisition (for live imaging), and details of objectives (magnification, numerical aperture) for all the different kinds of imaging experiments described in methods.
- Please modify the data availability statement to read, "All raw data generated in this study will be made available upon request"
- Please do a thorough spell and grammar check on the manuscript document
- In the abstract, we request you to modify the sentence in line 32. Our suggestion is as follows, "We show that upon DNA replication stress (a condition typically associated with CIN), Wnt10b acts to prevent increased microtubule dynamics from S phase until mitosis, thereby ensuring faithful chromosome segregation"
- Please be sure that the authorship listing and order are correct and match between the system and the manuscript file
- Please add your main and supplementary figure legends to the main manuscript text after the references section
- We encourage you to revise the figure legend for Figure S5 such that the figure panels are introduced in alphabetical order
- Please add callouts for Figures S5D and S6A-B to your main manuscript text
- Please add the X and Bluesky handles of your host institute/organization, as well as your own and/or one of the authors in our system

A. FINAL FILES:

-- Summary blurb (enter in submission system): A short text summarizing in a single sentence the study (max. 200 characters including spaces). This text is used in conjunction with the titles of papers, hence should be informative and complementary to

the title. It should describe the context and significance of the findings for a general readership; it should be written in the present tense and refer to the work in the third person. Author names should not be mentioned.

B. MANUSCRIPT ORGANIZATION AND FORMATTING:

Sincerely,

Sarita Hebbar, PhD
Scientific Editor
Life Science Alliance
<http://www.lsajournal.org>

Reviewer #2 (Comments to the Authors (Required)):

This manuscript for the first time reports the novel involvement of Wnt10b signalling in colorectal cancer cells. Wnt10b emerges as a novel factor, which ensures proper chromosome segregation and is essential for the suppression of DNA damage and aneuploidy. Furthermore, multiple independent assays have been performed and the results nicely highlight the overarching relevance and requirement for Wnt10b signalling for ensuring chromosomal stability. The authors have carefully addressed all the points raised by each of the reviewers and I emphatically support the acceptance and publication of this manuscript.

Reviewer #3 (Comments to the Authors (Required)):

The authors have made considerable efforts to address my previous concerns and have responded thoroughly to the majority of the points raised. I have no major concerns remaining.

Minor points and suggestions:

Figure S4B: The inter-origin distance analysis (previously Figure S3A) has been removed. I suggest that this information be kept, as it provides insights into replication dynamics.

Regarding the potential modulation of replication stress by Wnt signaling, even subtle effects on replication dynamics should not be disregarded, as they could partially account for the observed impact of Wnt10b on chromosomal breaks. The authors appropriately discuss the potential role of Wnt in replication fork integrity or repair in the Discussion, which enhances rather than detracts from their central message.

Life Science Alliance Manuscript #LSA-2025-03295R (Haas et al.)
Corresponding author: Holger Bastians, University Medical Center Göttingen, Germany

Dear Dr. Hebbbar

Please find below our point-by-point response to your remaining editorial comments:

-We encourage the author to head to a minor suggestion for Figure S4B (previously Figure S3A) from Reviewer 3.

As suggested, we re-included the omitted figure, which is now presented as Figure S4A (fork speed and origin firing in HCT116 cells) and Figure S4C (fork speed and origin firing in RPE1-hTert cells). Thus, the figure S4 has now 4 instead of 3 panels (S4A-D) and are described in the result section (pages 8 and 9).

-Please provide information on temperature of stage/slide during image acquisition (for live imaging), and details of objectives (magnification, numerical aperture) for all the different kinds of imaging experiments described in methods.

As suggested, we now include the information on the used objectives and temperatures in the method section: *Measurements of microtubule plus-end growth rates* (page 18), *Detection of chromosome missegregation* (page 20), *Detection of chromosomal breaks* (page 20) and *Molecular DNA combing* (page 21).

-Please modify the data availability statement to read, "All raw data generated in this study will be made available upon request"

We changed the sentence as suggested (page 21).

-Please do a thorough spell and grammar check on the manuscript document

We performed a thorough spell and grammar check.

-In the abstract, we request you to modify the sentence in line 32. Our suggestion is as follows, "We show that upon DNA replication stress (a condition typically associated with CIN), Wnt10b acts to prevent increased microtubule dynamics from S phase until mitosis, thereby ensuring faithful chromosome segregation"

We changed the sentence as suggested (page 2).

-Please be sure that the authorship listing and order are correct and match between the system and the manuscript file

We adjusted the author names accordingly in the submission system (A. Haas is first author, H. Bastians is last and corresponding author).

-Please add your main and supplementary figure legends to the main manuscript text after the references section

We added the supplementary figure legends into the main text file.

-We encourage you to revise the figure legend for Figure S5 such that the figure panels are introduced in alphabetical order

We renamed the panels in Figure S5 to panels S5A, S5B and S5C.

-Please add callouts for Figures S5D and S6A-B to your main manuscript text

We corrected this mistake in the main text. Figure S5D is not existing, instead its S5C. S6A,B are now mentioned on page 10.

-Please add the X and Bluesky handles of your host institute/organization, as well as your own and/or one of the authors in our system

Our institution is not supporting X or Bluesky.

Sincerely,

Holger Bastians (corresponding author)

August 8, 2025

RE: Life Science Alliance Manuscript #LSA-2025-03295RR

Prof. Holger Bastians
Universitätsmedizin Göttingen
Institute of Molecular Oncology
Grisebachstrasse 8
Goettingen 37077
Germany

Dear Dr. Bastians,

Thank you for submitting your Research Article entitled "Wnt10b signaling regulates replication stress-induced chromosomal instability in human cancer".

It is a pleasure to let you know that your manuscript is now accepted for publication in Life Science Alliance. Congratulations on this interesting work.

DISTRIBUTION OF MATERIALS:

Again, congratulations on a very nice paper. I hope you found the review process to be constructive and are pleased with how the manuscript was handled editorially. We look forward to future exciting submissions from your lab.

Sincerely,

Sarita Hebbar, PhD
Scientific Editor
Life Science Alliance
<http://www.lsajournal.org>